# Graph Neural Network Bandits

**Parnian Kassraie**
ETH Zurich
pkassraie@ethz.ch

**Andreas Krause**
ETH Zurich
krausea@ethz.ch

**Ilija Bogunovic**
University College London
i.bogunovic@ucl.ac.uk

## Abstract

We consider the bandit optimization problem with the reward function defined over graph-structured data. This problem has important applications in molecule design and drug discovery, where the reward is naturally invariant to graph permutations. The key challenges in this setting are scaling to large domains, and to graphs with many nodes. We resolve these challenges by embedding the permutation invariance into our model. In particular, we show that graph neural networks (GNNs) can be used to estimate the reward function, assuming it resides in the Reproducing Kernel Hilbert Space of a permutation-invariant additive kernel. By establishing a novel connection between such kernels and the graph neural tangent kernel (GNTK), we introduce the first GNN confidence bound and use it to design a phased-elimination algorithm with sublinear regret. Our regret bound depends on the GNTK's maximum information gain, which we also provide a bound for. While the reward function depends on all $N$ node features, our guarantees are *independent* of the number of graph nodes $N$. Empirically, our approach exhibits competitive performance and scales well on graph-structured domains.

## 1 Introduction

Contemporary bandit optimization approaches consider problems on large or continuous domains and have successfully been applied to a significant number of machine learning and real-world applications, e.g., in mobile health, environmental monitoring, economics, and hyperparameter tuning, to name a few. The main idea behind them is to exploit the correlations between the rewards of "similar" actions. This in turn, has resulted in increasingly rich models of reward functions (e.g., in linear and kernelized bandits [37, 12]), including several recent attempts to harness deep neural networks for bandit tasks (see, e.g, [45, 29]). A vast majority of previous works only focus on standard input domains and obtaining theoretical regret bounds.

Learning on graph-structured data, such as molecules or biological graph representations, requires designing sequential methods that can effectively exploit the structure of graphs. Consequently, graph neural networks (GNNs) have received attention as a rapidly expanding class of machine learning models. They deem remarkably well-suited for prediction tasks in applications such as designing novel materials [20], drug discovery [22], structure-based protein function prediction [16], etc. This rises the question of how to bridge the gap, and design bandit optimization algorithms on graph-structured data that can exploit the power of graph neural networks to approximate a graph reward.

In this paper, we consider bandit optimization over graphs and propose to employ graph neural networks with one convolutional layer to estimate the unknown reward. To scale to *large* graph domains (both in the number of graphs and number of nodes), we propose practical structural assumptions to model the reward function. In particular, we propose to use *permutation invariant additive* kernels. We show a novel connection between such kernels and the *graph neural tangent kernel* (GNTK) that we define in Section 3. Our main result are *GNN confidence bounds* that can be readily used in sequential decision-making algorithms to achieve sublinear regret bounds (see Section 4.3).

36th Conference on Neural Information Processing Systems (NeurIPS 2022).

**Related Work.** Our work extends the rich toolbox of methods for kernelized bandits and Bayesian optimization (BO) that work under the norm bounded Reproducing Kernel Hilbert Space assumption [37, 14, 42, 12]). The majority of these methods are designed for general Euclidean domains and rely on kernelized confidence sets to select which action to query next. The exception is [43], that consider the spectral setting in which the reward function is a linear combination of the eigenvectors of the graph Laplacian and the bandit problem is defined over nodes of a single graph. In contrast, our focus is on optimizing over graph domains (i.e., set of graphs), and constructing confidence sets that can quantify the uncertainty of graph neural networks estimates.

This work contributes to the literature on *neural bandits*, in which a fully-connected [45, 44, 19], or a single hidden layer convolutional network [25] is used to estimate the reward function. These works provide sublinear cumulative regret bounds in their respective settings, however, when applied directly to graph features (as we demonstrate in Section 4.3), these approaches do not scale well with the number of graph nodes.

Due to its important applications in molecule design, sequential optimization on graphs has recently received considerable attention. For example, in [28], the authors propose a kernel to capture similarities between graphs, and at every step, select the next graph through a kernelized random walk. Other works (e.g., [17, 18, 23, 38]) encode graph representations to the (continuous) latent space of a variational autoencoder and perform BO in the latent space. While practically relevant for discovering novel molecules with optimized properties, these approaches lack theoretical guarantees and deem computationally demanding.

A primary focus in our work is on embedding the natural structure of the data, i.e., permutation invariance, into the reward model. This is inspired by the works of [6, 32] that consider invariances in kernel-based supervised learning. Consequently, the graph neural tangent kernel plays an integral role in our theoretical analysis. Du et al. [15] provide a recursive expression for the tangent kernel of a GNN, without showing that the obtained expression is the limiting tangent kernel as defined in Jacot et al. [21] (i.e., as in Eq. (3)). In contrast, we analyze the learning dynamics of the GNN and properties of the GNTK by exploiting the connection between the structure of a graph neural network and that of a neural network (in Section 3). We recover that the graph neural tangent kernel also encodes additivity. Additive models for bandit optimization have been previously studied in [24] and [35], however, these works only focus on Euclidean domains and standard base kernels.

Finally, we build upon the recent literature on elimination-based algorithms that make use of maximum variance reduction sampling [13, 8, 7, 9, 40, 30]. One of our proposed algorithms, GNN-PE, employs a phased elimination strategy together with our GNN confidence sets.

**Main Contributions.** We introduce a bandit problem over graphs and propose to capture prior knowledge by modeling the unknown reward function using a permutation invariant additive kernel. We establish a key connection between such kernel assumptions and the graph neural tangent kernel (Proposition 3.2). By exploiting this connection, we provide novel statistical confidence bounds for the graph neural network estimator (Theorem 4.2). We further prove that a phased elimination algorithm that uses our GNN-confidence bounds (GNN-PE) achieves sublinear regret (Theorem 4.3). Importantly, our regret bound scales favorably with the number of graphs and is *independent* of the number of graph nodes (see Table 1). Finally, we empirically demonstrate that our algorithm consistently outperforms baselines across a range of problem instances.

## 2 Problem Statement

We consider a bandit problem where the learner aims to optimize an *unknown* reward function via sequential interactions with a stochastic environment. At every time step $t \in \{1, \ldots, T\}$, the learner selects a graph $G_t$ from a graph domain $\mathcal{G}$ and observes a noisy reward $y_t = f^*(G_t) + \epsilon_t$, where $f^* : \mathcal{G} \to \mathbb{R}$ is the reward function and $\epsilon_t$ is i.i.d. zero-mean sub-Gaussian noise with known variance proxy $\sigma^2$. Over a time horizon $T$, the learner seeks a small *cumulative* regret $R_T = \sum_{t=1}^{T} f^*(G^*) - f^*(G_t)$, where $G^* \in \arg\max_{G \in \mathcal{G}} f^*(G)$. The aim is to attain regret that is *sublinear* in $T$, meaning that $R_T/T \to 0$ as $T \to \infty$, which implies convergence to the optimal graph. As an example application, consider drug or material design, where molecules may be represented with graph structures (e.g., from SMILES representations [1]) and the reward $f^*(G)$ can correspond to an unknown molecular property of interest, e.g., atomization energy. Evaluating such properties typically requires running

costly simulations or experiments with noisy outcomes. To identify the most promising candidate, e.g., the molecule with the highest atomization energy, molecules are sequentially recommended for testing and the goal is to find the optimal molecule with the least number of evaluations.

**Graph Domain.** We assume that the domain $\mathcal{G}$ is a finite set of undirected graphs with $N$ nodes.[1] Without exploiting structure, standard bandit algorithms (e.g., [3]) cannot generalize across graphs, and their regret linearly depends on $|\mathcal{G}|$. To capture the structure, we consider reward functions depending on features associated with the graph nodes. Similar to Du et al. [15], we associate each node $j \in [N]$ with a feature vector $\boldsymbol{h}_{G,j} \in \mathbb{R}^d$, for every graph $G \in \mathcal{G}$. We use $\boldsymbol{h}_G = (\boldsymbol{h}_{G,j})_{j=1}^N \in \mathbb{R}^{Nd}$ to denote the concatenated vector of all node features, and $\mathcal{N}(j)$ as the neighborhood of node $j$, including itself. We define the aggregated node feature $\bar{\boldsymbol{h}}_{G,j} = \sum_{i \in \mathcal{N}(j)} \boldsymbol{h}_{G,i} / \|\sum_{i \in \mathcal{N}(j)} \boldsymbol{h}_{G,i}\|_2$ as the normalized sum of the neighboring nodes' features. Similarly, $\bar{\boldsymbol{h}}_G \in \mathbb{R}^{Nd}$ denotes the aggregated features, stacked across all nodes. Lastly, we let $P_N$ be the group of all permutations of length $N$, and use $c \cdot G$ to denote a permuted graph, where a permutation $c \in P_N$ is a bijective mapping from $\{1, \ldots, N\}$ onto itself. Permuting the nodes of a graph $c \cdot G$ produces a permuted feature vector $\boldsymbol{h}_{c \cdot G} := (\boldsymbol{h}_{G,c(j)})_{j=1}^N$, and the same holds for the aggregated features $\bar{\boldsymbol{h}}_{c \cdot G}$.

**Reward Model.** Practical graph optimization problems, such as drug discovery and materials optimization often do not depend on how the graphs' nodes in the dataset are ordered. We incorporate this structural prior into modeling the reward function, and consider functions that are *invariant to node permutations*. We assume that $f^*$ depends on the graph only through the aggregated node features and gives the same reward for all permutations of a graph, i.e., $f^*(c \cdot G) = f^*(G)$, for any $G \in \mathcal{G}$ and $c \in P_N$. To guarantee such an invariance, we assume that the reward belongs to the reproducing kernel Hilbert space (RKHS) corresponding to a permutation invariant kernel

$$\bar{k}(G, G') = \frac{1}{|P_N|^2} \sum_{c,c' \in P_N} k(\bar{\boldsymbol{h}}_{c \cdot G}, \bar{\boldsymbol{h}}_{c' \cdot G'}),$$

where $k$ can be any kernel defined on graph representations $\bar{\boldsymbol{h}}_G$. This assumption further restricts the hypothesis space to permutation invariant functions defined on $Nd$–dimensional vector representations of graphs. This is due to the reproducing property of the RKHS which allows us to write $f(G) = \langle f, \bar{k}(G, \cdot) \rangle = \langle f, \bar{k}(c \cdot G, \cdot) \rangle = f(c \cdot G)$. To make progress when optimizing over graphs with a *large* number of nodes $N$, we assume that $k$ decomposes additively over node features, i.e.,

$$k(\bar{\boldsymbol{h}}_G, \bar{\boldsymbol{h}}_{G'}) = \frac{1}{N} \sum_{j=1}^N k^{(j)}(\bar{\boldsymbol{h}}_{G,j}, \bar{\boldsymbol{h}}_{G',j}).$$

Thereby, we obtain an *additive* graph kernel that is *invariant* to node permutations:

$$\bar{k}(G, G') = \frac{1}{|P_N|^2} \sum_{c,c' \in P_N} \frac{1}{N} \sum_{j=1}^N k^{(j)}(\bar{\boldsymbol{h}}_{G,c(j)}, \bar{\boldsymbol{h}}_{G',c'(j)}). \tag{1}$$

For an arbitrary choice of $k^{(j)}$, calculating $\bar{k}$ requires a costly sum over $(N!)^2$ operands, since $|P_N| = N!$. In Section 3, we select a base kernel for which the sum can be reduced to $N^2$ terms. We are now in a position to state our main assumption. We assume that $f^*$ belongs to the RKHS of $\bar{k}$ and has a $B$-bounded RKHS norm. The norm-bounded RKHS regularity assumption is typical in the kernelized and neural bandits literature [37, 12, 45, 25]. Note that Eq. (1) only puts a structural prior on the kernel function, i.e., it describes the generic form of an additive permutation invariant graph kernel. Specifying the base kernels $k^{(j)}$ determines the representation power of $\bar{k}$. The smoother the base kernels are, the less complex the RKHS of $\bar{k}$ will be. In Section 3, we set the base kernels $k^{(j)}$ such that $\bar{k}$ becomes the expressive graph neural tangent kernel.

## 3 Graph Neural Networks

Graph neural networks are effective models for learning complex functions defined on graphs. As in Du et al. [15], we consider graph networks that have a *single* graph convolutional layer and $L$

---

[1]This assumption is for ease of exposition. Graphs with fewer than $N$ nodes can be treated by adding auxiliary nodes with no features that are disconnected from the rest of the graph.

fully-connected ReLU layers of equal width $m$. Such a network $f_{\mathrm{GNN}}(G; \boldsymbol{\theta}) : \mathcal{G} \to \mathbb{R}$ may be recursively defined as follows:

$$f^{(1)}(\bar{\boldsymbol{h}}_{G,j}) = \boldsymbol{W}^{(1)}\bar{\boldsymbol{h}}_{G,j},$$

$$f^{(l)}(\bar{\boldsymbol{h}}_{G,j}) = \sqrt{\frac{2}{m}}\boldsymbol{W}^{(l)}\sigma_{\mathrm{relu}}\big(f^{(l-1)}(\bar{\boldsymbol{h}}_{G,j})\big) \in \mathbb{R}^m, 1 < l \leq L \tag{2}$$

$$f_{\mathrm{GNN}}(G; \boldsymbol{\theta}) = \frac{1}{N}\sum_{j=1}^{N}\sqrt{2}\boldsymbol{W}^{(L+1)}\sigma_{\mathrm{relu}}\big(f^{(L)}(\bar{\boldsymbol{h}}_{G,j})\big),$$

where $\boldsymbol{\theta} := (\boldsymbol{W}^{(i)})_{i \leq L+1}$ is initialized randomly with standard normal i.i.d. entries, and $\sigma_{\mathrm{relu}}(\boldsymbol{x}) := \max(\boldsymbol{0}, \boldsymbol{x})$. The network operates on aggregated node features $\bar{\boldsymbol{h}}_{G,j}$ as typical in Graph Convolutional Networks [27]. For convenience, we assume that at initialization $f_{\mathrm{GNN}}(G; \boldsymbol{\theta}^0) = 0$, for all $G \in \mathcal{G}$, similar to [25, 45]. This assumption can be fulfilled without loss of generality, with a similar treatment as in [25, Appendix B.2].

**Embedded Invariances.** In this work, we use graph neural networks to estimate the unknown reward function $f^*$. This choice is motivated by the expressiveness of the GNN, the fact that it scales well with graph size, and particularly due to the invariances embedded in its structure. We observe that the graph neural network $f_{\mathrm{GNN}}$ is invariant to node permutations, i.e., for all $G \in \mathcal{G}$ and $c \in P_N$,

$$f_{\mathrm{GNN}}(G; \boldsymbol{\theta}) = f_{\mathrm{GNN}}(c \cdot G; \boldsymbol{\theta}).$$

The key step to show this property is proving that $f_{\mathrm{GNN}}$ can be expressed as an additive model of $L$-layer fully-connected ReLU networks,

$$f_{\mathrm{GNN}}(G; \boldsymbol{\theta}) = \frac{1}{N}\sum_{j=1}^{N}f_{\mathrm{NN}}(\bar{\boldsymbol{h}}_{G,j}; \boldsymbol{\theta}),$$

where $f_{\mathrm{NN}}$ has a similar recursive definition as $f_{\mathrm{GNN}}$ (see Equation A.1). The above properties are formalized in Lemma A.1 and Lemma A.2.

**Lazy (NTK) Regime.** We initialize and train $f_{\mathrm{GNN}}$ in the well-known lazy regime [11]. In this initialization regime, when the width $m$ is large, training with gradient descent using a small learning rate causes little change in the network's parameters. Let $\boldsymbol{g}_{\mathrm{GNN}}(G, \boldsymbol{\theta}) = \nabla_{\boldsymbol{\theta}}f_{\mathrm{GNN}}(G, \boldsymbol{\theta})$ denote the gradient of the network. It can be shown that during training, for all $G \in \mathcal{G}$, the network remains close to $f_{\mathrm{GNN}}(G, \boldsymbol{\theta}^0) + \boldsymbol{g}_{\mathrm{GNN}}^T(G, \boldsymbol{\theta}^0)(\boldsymbol{\theta} - \boldsymbol{\theta}^0)$, that is, its first order approximation around initialization parameters $\boldsymbol{\theta}^0$. Training this linearized model with a squared error loss is equivalent to kernel regression with a *tangent kernel* $\tilde{k}_{\mathrm{GNN}}(G, G') := \boldsymbol{g}_{\mathrm{GNN}}^T(G; \boldsymbol{\theta}^0)\,\boldsymbol{g}_{\mathrm{GNN}}(G'; \boldsymbol{\theta}^0)$. For networks of finite width, this kernel function is random since it depends on the random network parameters at initialization. We show in Proposition 3.1, that in the infinite width limit, the tangent kernel converges to a deterministic kernel. This proposition introduces the Graph Neural Tangent Kernel as the limiting kernel, and links it to the Neural Tangent Kernel ([21], also defined in Appendix A).

**Proposition 3.1.** *Consider any two graphs $G$ and $G'$ with $N$ nodes and $d$-dimensional node features. In the infinite width limit, the tangent kernel $\tilde{k}_{\mathrm{GNN}}(G, G')/m$ converges to a deterministic kernel,*

$$k_{\mathrm{GNN}}(G, G') := \lim_{m \to \infty} \tilde{k}_{\mathrm{GNN}}(G, G')/m. \tag{3}$$

*which we refer to as the Graph Neural Tangent Kernel (GNTK). Moreover,*

$$k_{\mathrm{GNN}}(G, G') = \frac{1}{N^2}\sum_{j,j'=1}^{N}k_{\mathrm{NN}}(\bar{\boldsymbol{h}}_{G,j}, \bar{\boldsymbol{h}}_{G',j'}) \tag{4}$$

*where $k_{\mathrm{NN}} : \mathbb{S}^{d-1} \times \mathbb{S}^{d-1} \to \mathbb{R}$ is the Neural Tangent Kernel.*

The proof is given in Appendix A.1. We note that $\bar{\boldsymbol{h}}_{G,j}$ lies on the $d$-dimensional sphere, since the aggregated node features are normalized. The NTK is bounded by 1 for any two points on the sphere [5]. Therefore, Proposition 3.1 implies that the GNTK is also bounded, i.e., $k_{\mathrm{GNN}}(G, G') \leq 1$ for any $G, G' \in \mathcal{G}$. This proposition yields a kernel which captures the behaviour of the lazy GNN. While defined on graphs with $Nd$ dimensional representations, the effective input domain of this

kernel is $d$-dimensional. This advantage directly stems from the additive construction of the GNTK. The next proposition uncovers the embedded structure of the GNTK by showing a novel connection between the GNTK and $\bar{k}$, the permutation invariant additive kernel from Eq. (1). The proof is presented in Appendix A.1.

**Proposition 3.2.** *Consider $\bar{k}$ from Eq. (1), where for every $1 \leq j \leq N$ the base kernel $k^{(j)}$ is set to be equal to $k_{\mathrm{NN}}$,*

$$\bar{k}(G, G') = \frac{1}{|P_N|^2} \sum_{c,c' \in P_N} \frac{1}{N} \sum_{j=1}^{N} k_{\mathrm{NN}}(\bar{\boldsymbol{h}}_{G,c(j)}, \bar{\boldsymbol{h}}_{G',c'(j)}).$$

*Then the permutation invariant additive kernel and the GNTK are identical, i.e., for all $G, G' \in \mathcal{G}$,*

$$\bar{k}(G, G') = k_{\mathrm{GNN}}(G, G').$$

This result implies that $k_{\mathrm{GNN}}$ inherits the favorable properties of the permutation invariant additive kernel class. Hence, functions residing in $\mathcal{H}_{\mathrm{GNN}}$, the RKHS of $k_{\mathrm{GNN}}$, are additive, invariant to node permutations, and act on $G$ through its aggregated node features. While we use the GNTK as an analytical tool, this kernel can be of independent interest in kernel methods over graph domains. In particular, calculating $k_{\mathrm{GNN}}$ requires significantly fewer operations compared to a kernel $\bar{k}$ with an arbitrary choice of $k^{(j)}$, for which calculating $\bar{k}(G, G')$ requires super-exponentially many operations in $N$ (See Eq. (1)). In contrast, due to the decomposition in Eq. (4), calculating $k_{\mathrm{GNN}}$ only costs a quadratic number of summations.

## 4 GNN Bandits

The bandit literature is rich with algorithms that effectively balance exploration and exploitation to achieve sublinear regret. Two components are common in kernelized bandit optimization algorithms. The *maximum information gain*, for characterizing the worst-case complexity of the learning problem [37, 24, 12, 40]; and *confidence sets*, for quantifying the learner's uncertainty [4, 39, 36, 12, 31]. Our first main result is an upper bound for the maximum information gain when the hypothesis space is $\mathcal{H}_{\mathrm{GNN}}$ (Theorem 4.1). We then propose valid confidence sets that utilize GNNs in Theorem 4.2. These theorems may be of independent interest, as they can be used towards bounding the regret for a variety of bandit algorithms on graphs. Lastly, we introduce the GNN-PE algorithm, together with its regret guarantee.

### 4.1 Information Gain

In bandit tasks, the learner seeks actions that give a large reward while, at the same time, provide information about the unknown reward function. The speed of learning about $f^*$ is commonly quantified via the maximum information gain. Assume that the learner chooses a sequence of actions $(G_1, \ldots, G_T)$ and observes noisy rewards, where the noise is i.i.d. and drawn from a zero-mean sub-Gaussian distribution with a variance proxy $\lambda$. The information gain of this sequence calculated via the GNTK is

$$I(G_1, \ldots, G_T; k_{\mathrm{GNN}}) = \frac{1}{2} \log \det(\boldsymbol{I} + \lambda^{-1} \boldsymbol{K}_{\mathrm{GNN},T})$$

with the kernel matrix $\boldsymbol{K}_{\mathrm{GNN},T} = [k_{\mathrm{GNN}}(G_i, G_j)]_{i,j \leq T}$. The maximum information gain (MIG) [37] is then defined as:

$$\gamma_{\mathrm{GNN},T} = \max_{\substack{(G_1, \cdots, G_T) \\ \forall t: G_t \in \mathcal{G}}} I(G_1, \cdots, G_T; k_{\mathrm{GNN}}). \tag{5}$$

In Section 4.3, we express regret bounds in terms of this quantity, as common in kernelized and neural bandits. In Theorem 4.1 we obtain a data-independent bound on the MIG. The proof is given in Appendix B.

**Theorem 4.1** (GNTK Information Gain Bound). *Suppose the observation noise is i.i.d., and drawn from a zero-mean sub-Gaussian distribution, and the input domain is $\mathcal{G}$. Then the maximum information gain associated with $k_{\mathrm{GNN}}$ is bounded by*

$$\gamma_{\mathrm{GNN},T} = \mathcal{O}\left(T^{\frac{d-1}{d}} \log^{\frac{1}{d}} T\right).$$

We observe that the obtained MIG bound *does not* depend on $N$ the number of nodes in the graphs. To highlight this advantage, we compare Theorem 4.1 to the equivalent bound for the vanilla neural tangent kernel which ignores the graph structure. We consider the neural tangent kernel that operates on graphs through the $Nd$-dimensional vector of aggregated node features $\bar{h}_G$,

$$\kappa_{\mathrm{NN}}(G, G') = \kappa_{\mathrm{NN}}\left(\frac{\bar{h}_G}{N}, \frac{\bar{h}_{G'}}{N}\right). \tag{6}$$

For $\kappa_{\mathrm{NN}}$ the maximum information gain scales as $\gamma_{\mathrm{NN},T} = \mathcal{O}(T^{(Nd-1)/Nd} \log^{1/Nd} T)$, where $N$ appears in the exponent [25]. This results in poor scalability with graph size in the bandit optimization task, as we further demonstrate in Section 4.3. Table 1 summarizes this comparison.

## 4.2 Confidence Sets

Quantifying the uncertainty over the reward helps the learner to guide exploration and balance it against exploitation. Confidence sets are an integral tool for uncertainty quantification. Conditioned on the history $H_{t-1} = (G_i, y_i)_{i<t}$, for any $G \in \mathcal{G}$, the set $\mathcal{C}_{t-1}(G, \delta)$ defines an interval to which $f^*(G)$ belongs with a high probability such that,

$$\mathbb{P}\left(\forall G \in \mathcal{G} : f^*(G) \in \mathcal{C}_{t-1}(G, \delta)\right) \geq 1 - \delta. \tag{7}$$

An approach common to the kernelized bandit literature is to construct sets of the form $\mathcal{C}_{t-1}(G, \delta) = [\mu_{t-1}(G) \pm \beta_t \sigma_{t-1}(G)]$ where $\beta_t$ depends on the confidence level $\delta$. The center of the interval, characterized by $\mu_{t-1}(\cdot)$, is the estimate of the reward, and the width $\beta_t \sigma_{t-1}(\cdot)$, reflects the uncertainty. In this work, we utilize GNNs for construction of such sets. To this end, we train a graph neural network to estimate the reward. We use the gradient of this network at initialization to approximate the uncertainty over the reward, as in [45]. Let $f_{\mathrm{GNN}}(G; \theta_{t-1}^{(J)})$ be the GNN trained with gradient descent for $J$ steps and by using learning rate $\eta$ on the loss

$$\mathcal{L}(\theta) = \frac{1}{t} \sum_{i<t} \left(f_{\mathrm{GNN}}(G_i, \theta) - y_i\right)_2^2 + m\lambda \|\theta - \theta^0\|_2^2,$$

where $\lambda$ is the regularization coefficient, and $\theta^0$ the network parameters at initialization. We propose confidence sets of the form

$$\mathcal{C}_{t-1}(G, \delta) = [\hat{\mu}_{t-1}(G) \pm \beta_t \hat{\sigma}_{t-1}(G)],$$

where the center and width of the set are calculated via,

$$\hat{\mu}_{t-1}(G) := f_{\mathrm{GNN}}(G; \theta_{t-1}^{(J)}),$$

$$\hat{\sigma}_{t-1}^2(G) := \frac{g_{\mathrm{GNN}}^T(G; \theta^0)}{\sqrt{m}} \left(\lambda I + \frac{1}{t} \sum_{i=1}^{t-1} \frac{g_{\mathrm{GNN}}(G_i; \theta^0) g_{\mathrm{GNN}}^T(G_i; \theta^0)}{m}\right)^{-1} \frac{g_{\mathrm{GNN}}(G; \theta^0)}{\sqrt{m}}. \tag{8}$$

Here $g_{\mathrm{GNN}}(G; \theta^0) = \nabla_\theta f_{\mathrm{GNN}}(G; \theta^0)$ denotes the gradient at initialization. Moreover, we use $\lambda_0 := \lambda_{\min}(K_{\mathrm{GNN}}) > 0$ to denote the minimum eigenvalue of the kernel matrix calculated for the entire domain, i.e., $K_{\mathrm{GNN}} = [k_{\mathrm{GNN}}(G, G')]_{G,G' \in \mathcal{G}}$. Theorem 4.2 shows that this construction gives valid confidence intervals, i.e., it satisfies Eq. (7), when the reward function lies in $\mathcal{H}_{\mathrm{GNN}}$ and has a bounded RKHS norm.

**Theorem 4.2** (GNN Confidence Bound). *Set $\delta \in (0, 1)$. Suppose $f^* \in \mathcal{H}_{k_{\mathrm{GNN}}}$ with a bounded norm $\|f^*\|_{k_{\mathrm{GNN}}} \leq B$. Assume that the random sequences $(G_i)_{i<t}$ and $(\epsilon_i)_{i<t}$ are statistically independent. Let the width $m = \mathrm{poly}\left(t, L, B, |\mathcal{G}|, \lambda, \lambda_0^{-1}, \log(N/\delta)\right)$, learning rate $\eta = C(Lm + m\lambda)^{-1}$ with some universal constant $C$, and $J \geq 1$. Then for all graphs $G \in \mathcal{G}$, with probability of at least $1 - \delta$,*

$$|f^*(G) - \hat{\mu}_{t-1}(G)| \lesssim \beta_t \hat{\sigma}_{t-1}(G),$$

*where $\hat{\mu}_{t-1}$ and $\hat{\sigma}_{t-1}$ are defined in Eq. (8) and*

$$\beta_t \approx \sqrt{2}B + \frac{\sigma}{\sqrt{\lambda}} \sqrt{2 \log 2|\mathcal{G}|/\delta}.$$

The "≈" notation in Theorem 4.2 omits the terms that vanish with $t$, i.e., are $o(1)$. An exact version of the theorem without the aforementioned approximations is given in Appendix C.1.

| Setting | MIG Bound, $\gamma_T$ | Cumulative Regret (Phased Elimination) |
|---|---|---|
| Neural | $\mathcal{O}\left(T^{\frac{Nd-1}{Nd}} \log^{\frac{1}{Nd}} T\right)$ | $\tilde{\mathcal{O}}\left(T^{\frac{2Nd-1}{2Nd}} \log^{\frac{1}{2Nd}} T\right)$ |
| Graph Neural | $\mathcal{O}\left(T^{\frac{d-1}{d}} \log^{\frac{1}{d}} T\right)$ | $\tilde{\mathcal{O}}\left(T^{\frac{2d-1}{2d}} \log^{\frac{1}{2d}} T\right)$ |

Table 1: Summary of main bounds for the NN and GNN approaches. Here $T$ denotes the bandit horizon, $N$ the number of nodes in each graph, and $d$ the dimension of node features. The GNN guarantees are *independent* of $N$.

## 4.3 Bandit Optimization with Graph Neural Networks

The developed confidence sets can be used to assist the learner with controlling the growth of regret. In this section, we give a concrete example on how our GNN confidence sets (Equation 8) can be used by an algorithm to solve bandit optimization tasks on graphs.

We introduce GNN-Phased Elimination (GNN-PE; see Algorithm 1) that consists of episodes of pure exploration over a set of plausible maximizer graphs, similar to [7, 30]. Each episode is followed by an elimination step, that makes use of GNN confidence bounds to shrink the set of plausible maximizers. More formally, at step $t$ during an episode $e$, the learner selects actions via $G_{e,t} = \arg\max_{G \in \mathcal{G}_e} \hat{\sigma}_{e,t-1}(G)$, where $\mathcal{G}_e$ is the set of graphs that might maximize $f^*$ according to the learner's current knowledge. Once the episode is over after $T_e$ steps, the set $\mathcal{G}_e$ is updated to contain graphs that still have a chance of being a maximizer according to the confidence bounds $[\hat{\mu}_{e,T_e}(G) \pm \beta_{T_e} \hat{\sigma}_{e,T_e}(G)]$ where $\hat{\mu}_{e,T_e}$ and $\hat{\sigma}_{e,T_e}$ are only computed based on the points within episode $e$.

Theorem 4.3 shows that GNN-PE incurs a sublinear control over the cumulative regret. We provide the proof in Appendix C. We use $\tilde{\mathcal{O}}(\cdot)$ notation to hide $\mathrm{polylog}(T)$ factors.

**Theorem 4.3.** *Set $\delta \in (0, 1)$. Suppose $f^* \in \mathcal{H}_{k_{\mathrm{GNN}}}$ with a bounded norm $\|f^*\|_{k_{\mathrm{GNN}}} \leq B$. Let the width $m = \mathrm{poly}\left(t, L, B, |\mathcal{G}|, \lambda, \lambda_0^{-1}, \log(N/\delta)\right)$, learning rate $\eta = C(Lm + m\lambda)^{-1}$ with some universal constant $C$, and $J \geq 1$. Then with probability at least $1 - \delta$, GNN-PE satisfies*

$$R_T = \tilde{\mathcal{O}}\left(\sqrt{T\gamma_{T,\mathrm{GNN}}}\left(B + \frac{\sigma}{\sqrt{\lambda}}\sqrt{\log|\mathcal{G}|/\delta}\right)\right).$$

We can observe the benefit of working with a graph neural network by comparing the bound in Theorem 4.3 with the regret for a structure-agnostic algorithm. Recall $\kappa_{\mathrm{NN}}$ the vanilla NTK, defined over the concatenated feature vectors (Equation 6). For the sake of this comparison, we ignore the geometric structure and assume that $f^* \in \mathcal{H}_{\mathrm{NN}}$. Swapping out $k_{\mathrm{GNN}}$ for $\kappa_{\mathrm{NN}}$, and respectively the GNN with an NN as defined in Eq. (A.1), we obtain NN-PE, the neural network counterpart of GNN-PE. This algorithm accepts $Nd$-dimensional input vectors as actions. Similar to Theorem 4.3, we can show that NN-PE can satisfy a guarantee of $\mathcal{O}\left(T^{(2Nd-1)/2Nd} \log^{1/2Nd} T\right)$ for the regret. This bound suggests that as $N$ grows, finding the optimal graph can become more challenging for the learner. Working with $k_{\mathrm{GNN}}$ to encode the structure of the bandit problem, and consequently using the GNN to solve it, removes the dependency on $N$ in the exponent. This result is summarized in Table 1.

We provide some intuition on why working with a permutation invariant model is beneficial for bandit optimization on graphs. Confidence sets which are constructed for member of $\mathcal{H}_{\mathrm{NN}}$ are larger, and result in sub-optimal action selection. Further, training the neural network is a more challenging task, since permutation invariance is not hard coded in the network architecture and has to be learned from the data. This results in less accurate reward estimates. We refer the reader to Appendix A.3 for a more rigorous discussion. There we compare $\mathcal{H}_{\mathrm{GNN}}$ and $\mathcal{H}_{\mathrm{NN}}$, the hypothesis spaces corresponding to the two models, through the Mercer decomposition of their kernels.

## 5 Experiments

We create synthetic datasets which may be of independent interest and can be used for evaluating and benchmarking machine learning algorithms on graph domains. Each dataset is constructed from a finite graph domain together with a reward function. The domains are generated randomly and

differ in properties of the member graphs that influence the problem complexity, e.g., number of nodes and edge density. Each domain $\mathcal{G}_{p,N}$ consists of Erdős-Rényi random graphs, where each graph has $N$ nodes, and between each two nodes there exists an edge with probability $p$. The node features are i.i.d. $d = 10$ dimensional standard Gaussian vectors. We choose $N \in \{5, 20, 100\}$, $p \in \{0.05, 0.2, 0.95\}$, and thereby sample a total of 9 different domains each containing 10000 graphs. For instance, $\mathcal{G}_{0.05,5}$ denotes the domain with sparse and small graphs, while $\mathcal{G}_{0.95,100}$ is the domain of dense graphs with many nodes. For every domain, we sample a random reward function $f : \mathcal{G}_{p,N} \to \mathbb{R}$ that is invariant to node permutations. We use $\mathrm{GP}(0, k_{\mathrm{GNN}})$ as a prior, and sample $f$ from its posterior GP. The posterior is calculated using a small random dataset $(G_i, y_i)_{i \leq 5}$, where $y_i$ are drawn independently from $\mathcal{N}(0, 1)$ and $G_i$ are randomly chosen from $\mathcal{G}_{p,N}$. The corresponding dataset is then $D_{p,N} = \{(G_i, f(G_i)) \,|\, G_i \in \mathcal{G}_{p,N}\}$.

**Experiment Setup.** Every performance curve in the paper shows an average over 20 runs of the corresponding bandit problem, each with a different action set sampled from $\mathcal{G}_{p,N}$. The shaded areas in all figures show the standard error across runs. In all experiments, the reward is observed with a zero-mean Gaussian noise of variance $\sigma = 10^{-2}$. We always set width $m = 2048$ and layers $L = 2$, for every type of network architecture. Four algorithms appear in our experiments. In addition to our main algorithm GNN-PE, we introduce GNN-UCB, which selects actions via $G_t = \arg\max \hat{\mu}_{t-1}(G) + \beta_t \hat{\sigma}_{t-1}(G)$, the classic UCB policy based on the GNN confidence sets. The pseudo-code is given in Appendix D.3. NN-UCB, introduced by [45], is the neural counterpart of GNN-UCB, and NN-PE as discussed in Section 4.3. To configure these algorithms, we only tune $\lambda$ and $\beta = \beta_t$, and we do so by using the simplest dataset $D_{0.05,5}$. We find that the algorithms are not sensitive to domain configurations and work for all $D_{p,N}$ out of the box. Therefore, the same values for $\lambda$ and $\beta$ are used across all experiments. We include the complete result of our hyperparameter search in Figure 5.

**Lazy training.** We initialize the graph neural networks (and the NNs) in the lazy regime as described in Eq. (2) (and Eq. A.1). Training a network in this regime with gradient descent causes little change in the weights. Consequently, it is challenging to effectively train a lazy network in practice. Therefore, the stopping criterion for gradient descent plays a crucial role in achieving sublinear regret. Inaccurate estimation of the reward function disturbs the balance of exploration and exploitation, and leads the learner to poor optima. To prevent this issue, we devise a stopping criterion that depends on the history $H_{t-1}$, such that, as $t$ grows, the network is often trained for more gradient descent steps $J$. This criterion can be employed by any neural bandit algorithm and may be of independent practical interest. The details of training with gradient descent, stopping and batching are given in Appendix D.2.

**Regret Experiments.** We assess the performance of the algorithms on bandit optimization tasks over different domains. In Figure 1, we show the inference cumulative regret $\hat{R}_T = \sum_{t \leq T} f^*(G^*) - \max_{G \in \mathcal{G}} \hat{\mu}_{t-1}(G)$, for which we select graph domains with $N = 20$ nodes and edge probability $p = 0.2$. Figure 6 shows the regret for all dataset configurations. To verify scalability with $|\mathcal{G}|$, we run the algorithms on action sets of increasing size $|\mathcal{G}| \in \{200, 500, 1000\}$. Figure 1 presents the results: GNN-PE consistently outperforms the other methods. It is evident that the algorithms built with GNN confidence sets find the optimal graph, regardless of the size of the domain. The GNN algorithms exhibit competitive performance, and attain sublinear regret for all dataset configurations. The neural methods however, may fail to scale and find the optima in limited time.

**Scalability with Graph Size.** In Section 4.3, we argue that using a neural network which takes $\bar{h}_G \in \mathbb{R}^{Nd}$ as the input, causes the regret to grow with $\mathcal{O}(T^{(2Nd-1)/2Nd})$. The additive structure of the GNN, however, allows the learner to work on a $d$-dimensional domain, independent of graph size. Figure 2 reflects this behaviour. Fixing $p = 0.2$, and $|\mathcal{G}| = 200$, we run the algorithm over domains with two graph sizes $N \in \{20, 100\}$. GNN-PE achieves sublinear regret in both cases, and manages to find a global maxima within roughly the same number of steps. This is in contrast to NN-PE, which is more affected by increasing graph size. A similar comparison for all configurations and algorithms is plotted in Figure 7, and the same behaviour is observed across all settings: the performance of GNN methods scales well with $N$, while this is not the case for NN methods.

**Effect of Graph Density.** As a final observation, we discuss the effect of edge density. Consider a complete graph $G$ with $\binom{N}{2}$ edges. The neighborhoods are symmetric and the aggregated node features $\bar{h}_{G,i}$ are identical for all $i \leq N$. Permutations on this graph will not change the output of either $f_{\mathrm{GNN}}$ or $f_{\mathrm{NN}}$. Therefore, we expect that for dense graphs, i.e., large values of $p$, using a permutation invariant model comes with fewer benefits for the learner. This is opposed to when the graph is sparse

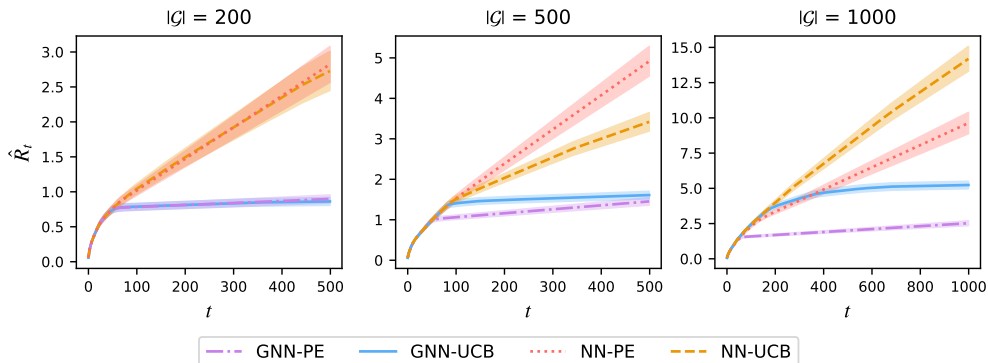

Figure 1: Regret $\hat{R}_T$ over a time horizon of 500 and 1000 steps with $N = 20$ and $p = 0.2$. GNN-PE consistently outperforms other algorithms and scales well with size of the action set $|\mathcal{G}|$.

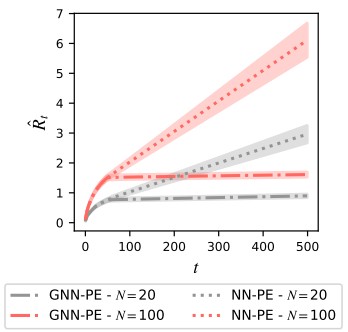

Figure 2: Increasing $N$ has little effect on GNN-PE.

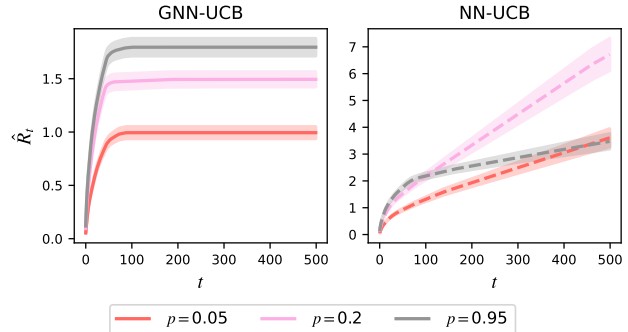

Figure 3: Increasing the edge density of the graphs reduces the performance gap between GNN-UCB and NN-UCB.

and the neighborhoods are asymmetric. To verify this conjecture, we fix $N = 20$, $|\mathcal{G}| = 200$, and run the algorithm over domains with graphs of different edge probability $p \in \{0.05, 0.2, 0.95\}$. Figure 3 shows that while GNN-UCB always achieves sublinear regret, it takes longer to find the optima when the graphs are more dense. NN-UCB however, improves as the edge probability $p$ grows, since, roughly put, the graphs in the domain are becoming invariant to permutations. Therefore Figure 3 confirms that the performance gap between the two method is reduced for graphs that are more dense. In Figure 8, we plot the effect of graph density for other dataset configurations and bandit algorithms. This behaviour is observed predominantly for the UCB algorithms.

## 6 Conclusion

We analyze the use of graph neural networks in bandit optimization tasks over large graph domains. The main takeaway is that encoding the natural structure of the environment into the model, reduces the complexity of the task for the learner. By selecting a kernel that embeds invariances, we introduce structure into the algorithm in a principled manner. Importantly, we propose key structural assumptions on the graph reward function and establish a novel connection between additive permutation invariant kernels and the GNTK. We construct valid graph neural network confidence sets, and use it to build a GNN bandit algorithm that achieves sublinear regret. While all node features contribute to the graph's reward, our bounds are independent of the number of nodes. This result holds for GNNs with a single convolutional layer and graphs with node feature representation. An immediate next step is to generalize this approach to other more complex graph neural network architectures and representations (e.g., by including information about graph edges) and investigating their effectiveness for bandit optimization. Our analysis opens up two avenues of future research. The proposed kernel and the graph confidence sets may be used in other algorithms for sequential decision-making tasks on graphs.

Additionally, our approach of embedding the environment's permutation invariant structure into the algorithm may inspire further work on structured bandit optimization in presence of invariances.

## Acknowledgments and Disclosure of Funding

We thank Jonas Rothfuss for his valuable suggestions regarding the experiments. We acknowledge Deepak Narayanan's effort on an earlier version of the code. We thank Nicolas Emmenegger and Scott Sussex for their thorough feedback, and lastly, we thank Alex Hägele for fruitful discussions regarding the writing. This research was supported by the European Research Council (ERC) under the European Union's Horizon 2020 research and Innovation Program Grant agreement no. 815943.

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
