# Supplementary Material: Graph Neural Network Bandits

## A The Neural Tangent Kernel and its Connection to the GNTK

Let $f(\boldsymbol{x}; \boldsymbol{\theta}) : \mathbb{R}^d \to \mathbb{R}$ be a fully-connected network, with $L$ hidden layers of equal width $m$, and ReLU activations, recursively defined as follows:

$$f^{(1)}(\boldsymbol{x}) = \boldsymbol{W}^{(1)}\boldsymbol{x},$$

$$f^{(l)}(\boldsymbol{x}) = \sqrt{\frac{2}{m}} \boldsymbol{W}^{(l)} \sigma_{\text{relu}}\big(f^{(l-1)}(\boldsymbol{x})\big) \in \mathbb{R}^m, \ 1 < l \le L \tag{A.1}$$

$$f_{\text{NN}}(\boldsymbol{x}; \boldsymbol{\theta}) = \sqrt{2} \boldsymbol{W}^{(L+1)} \sigma_{\text{relu}}\big(f^{(L)}(\boldsymbol{x})\big) \in \mathbb{R}.$$

The weights $\boldsymbol{W}^{(i)}$ are initialized to random matrices with standard normal i.i.d. entries, and $\boldsymbol{\theta}^0 = (\boldsymbol{W}^{(i)})_{i \le L+1}$. Consider the first order approximation of $f_{\text{NN}}(\boldsymbol{x}, \boldsymbol{\theta})$ around the initial parameters $\boldsymbol{\theta}^0$, i.e.,

$$\tilde{f}_{\text{NN}}(\boldsymbol{x}; \boldsymbol{\theta}) = \boldsymbol{g}_{\text{NN}}^T(\boldsymbol{x}, \boldsymbol{\theta}^0)(\boldsymbol{\theta} - \boldsymbol{\theta}^0),$$

since the network is defined to be zero at initialization. By considering a fixed dataset and a square loss, training with the linear model $\tilde{f}_{\text{NN}}(\boldsymbol{x}, \boldsymbol{\theta})$, is equivalent to regression with the *tangent kernel* [21], defined as

$$\tilde{k}_{\text{NN}}(\boldsymbol{x}, \boldsymbol{x}') = \boldsymbol{g}_{\text{NN}}^T(\boldsymbol{x}; \boldsymbol{\theta}^0)\boldsymbol{g}_{\text{NN}}(\boldsymbol{x}'; \boldsymbol{\theta}^0). \tag{A.2}$$

The tangent kernel is random since it depends on $\boldsymbol{\theta}^0$. Jacot et al. [21] show that in the infinite width limit, $\tilde{k}_{\text{NN}}(G, G')$ converges to a deterministic kernel, which they call the Neural Tangent Kernel (NTK),

$$\lim_{m \to \infty} \tilde{k}_{\text{NN}}(\boldsymbol{x}, \boldsymbol{x}')/m = k_{\text{NN}}(\boldsymbol{x}, \boldsymbol{x}').$$

The NTK satisfies the Mercer condition and has the following Mercer decomposition [5],

$$k_{\text{NN}}(\boldsymbol{x}, \boldsymbol{x}') = \sum_{r=0}^{\infty} \lambda_r \sum_{s=1}^{M(d,r)} Y_{s,r}(\boldsymbol{x}) Y_{s,r}(\boldsymbol{x}'), \tag{A.3}$$

where $\{Y_{s,r}\}_{s \le M(d,r)}$ form an orthonormal basis for $V(d,r)$ the space of degree-$r$ polynomials on $\mathbb{S}^{d-1}$. They eigenvalues $\lambda_r$ decay at a $r^{-d}$ rate [5]. The eigenfunction $Y_{s,r}$ is the $s$-th spherical harmonic polynomial of degree $r$, and $M(d,r) = \dim(V_{d,r})$ gives the total count of such polynomials, where

$$M(d,r) = \frac{2k+d-2}{r} \binom{r+d-3}{d-2}.$$

The NTK adopts a recursive definition (see Appendix A.2 ). Its properties and connections to infinite-width fully-connected networks are studied in detail [2, 5, 10].

### A.1 Properties of GNN and GNTK

We first note the connection between $f_{\text{GNN}}$ and $f_{\text{NN}}$.

**Lemma A.1** (GNN as sums of NNs)**.** *Consider* $f_{\text{GNN}}$ *the graph neural network defined in Eq. (2), and the feedforward network* $f_{\text{NN}}$ *as given in Eq. (A.1). Then,*

$$f_{\text{GNN}}(G, \boldsymbol{\theta}) = \frac{1}{N} \sum_{j=1}^{N} f_{\text{NN}}(\bar{\boldsymbol{h}}_{G,j}, \boldsymbol{\theta}).$$

**Proof of Lemma A.1.** According to Eq. (A.1), the two layer NN with width $m$ decomposes as:

$$f_{\text{NN}}(\boldsymbol{x}; \boldsymbol{\theta}) = \sqrt{2} \sum_{j=1}^{m} w_j^{(2)} \sigma_{\text{relu}}(\langle \boldsymbol{w}_j^{(1)}, \boldsymbol{x} \rangle), \tag{A.4}$$

where $\boldsymbol{w}_j^{(1)} \in \mathbb{R}^d$ are weights in the first layer and $w_j^{(2)} \in \mathbb{R}$ are weights in the second layer. Similarly, the two layer GNN (see Eq. (2)) is given by:

$$f_{\text{GNN}}(G; \boldsymbol{\theta}) = \frac{\sqrt{2}}{N} \sum_{i=1}^{N} \sum_{j=1}^{m} w_j^{(2)} \sigma_{\text{relu}}(\langle \boldsymbol{w}_j^{(1)}, \bar{\boldsymbol{h}}_{G,i} \rangle) \tag{A.5}$$

$$= \frac{1}{N} \sum_{i=1}^{N} f_{\text{NN}}(\bar{\boldsymbol{h}}_{G,i}; \boldsymbol{\theta}), \tag{A.6}$$

where Eq. (A.6) follows from Eq. (A.4). The relation in Eq. (A.6) holds trivially for arbitrary $L$. $\square$

We are now ready to show the permutation invariance property.

**Lemma A.2** (Geometric Invariance of $f_{\text{GNN}}$). *The graph neural network $f_{\text{GNN}}$ is invariant to node permutations, i.e., for all $G \in \mathcal{G}$ and $c \in P_N$,*

$$f_{\text{GNN}}(G; \boldsymbol{\theta}) = f_{\text{GNN}}(c \cdot G; \boldsymbol{\theta})$$

**Proof of Lemma A.2.** Consider any permutation $c \in P_N$. By Lemma A.1,

$$f_{\text{GNN}}(c \cdot G; \boldsymbol{\theta}) = \frac{1}{N} \sum_{j=1}^{N} f_{\text{NN}}(\bar{\boldsymbol{h}}_{G,c(j)}; \boldsymbol{\theta}) = \frac{1}{N} \sum_{i=1}^{N} f_{\text{NN}}(\bar{\boldsymbol{h}}_{G,i}; \boldsymbol{\theta}) = f_{\text{GNN}}(G; \boldsymbol{\theta}). \tag{A.7}$$

Since the summation over $f_{\text{NN}}(\bar{\boldsymbol{h}}_{G,c(j)}; \boldsymbol{\theta})$ for all $j$, contains the same terms as a sum over $f_{\text{NN}}(\bar{\boldsymbol{h}}_{G,i}; \boldsymbol{\theta})$ for all $i$. $\square$

We now prove that $k_{\text{GNN}}$ as defined in Section 2, is deterministic and can be written as a double sum of $k_{\text{NN}}$'s evaluated on $\boldsymbol{h}_{G,j}$ aggregated features of different nodes of the graph.

**Proof of Proposition 3.1.** For any $\boldsymbol{\theta}^0$ we first show that,

$$\tilde{k}_{\text{GNN}}(G, G') = \frac{1}{N^2} \sum_{j,j'=1}^{N} \tilde{k}_{\text{NN}}(\bar{\boldsymbol{h}}_{G,j}, \bar{\boldsymbol{h}}_{G',j'}), \tag{A.8}$$

where $\tilde{k}_{\text{NN}}(\cdot, \cdot)$ is from Eq. (A.2). Then, we take the $m \to \infty$ limit. Starting from the definition of $\tilde{k}_{\text{GNN}}$ (see Section 3) and by omitting $\boldsymbol{\theta}^0$ for simplicity of notation, we have:

$$\tilde{k}_{\text{GNN}}(G, G') = \boldsymbol{g}_{\text{GNN}}^T(G) \boldsymbol{g}_{\text{GNN}}(G')$$

$$\overset{\text{Lemma A.1}}{=} \left[ \sum_{j=1}^{N} \frac{1}{N} \boldsymbol{g}_{\text{NN}}^T(\bar{\boldsymbol{h}}_{G,j}) \right] \left[ \sum_{j'=1}^{N} \frac{1}{N} \boldsymbol{g}_{\text{NN}}(\bar{\boldsymbol{h}}_{G',j'}) \right]$$

$$= \frac{1}{N^2} \sum_{j,j'=1}^{N} \boldsymbol{g}_{\text{NN}}^T(\bar{\boldsymbol{h}}_{G,j'}) \boldsymbol{g}_{\text{NN}}^{(j)}(\bar{\boldsymbol{h}}_{G',j})$$

$$\overset{\text{A.2}}{=} \frac{1}{N^2} \sum_{j,j'=1}^{N} \tilde{k}_{\text{NN}}(\bar{\boldsymbol{h}}_{G,j}, \bar{\boldsymbol{h}}_{G',j'}).$$

The chain of equations above prove Eq. (A.8). Plugging in the definition of the GNTK, we obtain:

$$k_{\text{GNN}}(G, G') = \lim_{m \to \infty} \tilde{k}_{\text{GNN}}(G, G')/m$$

$$= \lim_{m \to \infty} \frac{1}{N^2} \sum_{j,j'=1}^{N} \tilde{k}_{\text{NN}}(\bar{\boldsymbol{h}}_{G,j}, \bar{\boldsymbol{h}}_{G',j'})/m$$

$$= \frac{1}{N^2} \sum_{j,j'=1}^{N} \lim_{m \to \infty} \tilde{k}_{\text{NN}}(\bar{\boldsymbol{h}}_{G,j}, \bar{\boldsymbol{h}}_{G',j'})/m$$

$$= \frac{1}{N^2} \sum_{j,j'=1}^{N} k_{\text{NN}}(\bar{\boldsymbol{h}}_{G,j}, \bar{\boldsymbol{h}}_{G',j'}),$$

where the second equality holds since $\tilde{k}_{\text{NN}}$ is continuous, and for continues functions, limit of finite sums is equal to sum of the limits. This concludes the proof. $\square$

***Proof of Proposition 3.2.*** From Proposition 3.1, we have

$$k_{\text{GNN}}(G, G') = \frac{1}{N^2} \sum_{j,j'=1}^{N} k_{\text{NN}}(\bar{\boldsymbol{h}}_{G,j}, \bar{\boldsymbol{h}}_{G,j'}).$$

It then suffices to show that $\bar{k}(G, G')$ (as defined in Eq. (1)) is equal to the right hand side of the above equation. Consider $P_N$ the set of permutations of $N$ elements. Every permutation $c \in P_N$ gives a mapping from $(1, \cdots, j, \cdots, N)$ to $(c(1), \cdots, c(j), \cdots, c(N))$, where $c(j) \in [N]$ denotes the element that is placed at the $j$-th position. We define a restricted set of permutations $P_{N|j \to i} = \{c \in P_N : c(j) = i\}$, such that

$$P_N = \bigcup_{i=1}^{N} P_{N|j \to i}. \tag{A.9}$$

Moreover, for any $1 \leq j \leq N$, $\{P_{N|j \to i}\}_{i=1}^{N}$ are disjoint sets and the cardinality of each restricted permutation set is

$$\left| P_{N|j \to i} \right| = (N-1)!, \tag{A.10}$$

which implies that the mapping $j \to i$ is repeated $(N-1)!$ times across the elements of $P_N$. Back to definition of $\bar{k}(G, G')$ we may decompose $P_N$ and write

$$\bar{k}(G, G') = \frac{1}{N!} \sum_{c' \in P_N} \left[ \frac{1}{N!} \sum_{c \in P_N} \frac{1}{N} \sum_{j=1}^{N} k_{\text{NN}}(\bar{\boldsymbol{h}}_{G,c(j)}, \bar{\boldsymbol{h}}_{G',c'(j)}) \right]$$

$$= \frac{1}{N!} \sum_{c' \in P_N} \left[ \frac{1}{N!} \frac{1}{N} \sum_{j=1}^{N} \sum_{c \in P_N} k_{\text{NN}}(\bar{\boldsymbol{h}}_{G,c(j)}, \bar{\boldsymbol{h}}_{G',c'(j)}) \right]$$

$$\overset{\text{Eq. (A.9)}}{=} \frac{1}{N!} \sum_{c' \in P_N} \left[ \frac{1}{N!} \frac{1}{N} \sum_{j=1}^{N} \sum_{i=1}^{N} \sum_{c \in P_{N|j \to i}} k_{\text{NN}}(\bar{\boldsymbol{h}}_{G,c(j)}, \bar{\boldsymbol{h}}_{G',c'(j)}) \right].$$

Now, by definition of $P_{N|i}$, we have $c(j) = i$ for all $c$ in this set. Therefore,

$$\bar{k}(G, G') = \frac{1}{N!} \sum_{c' \in P_N} \left[ \frac{1}{N!} \frac{1}{N} \sum_{j=1}^{N} \sum_{i=1}^{N} \sum_{c \in P_{N|j \to i}} k_{\text{NN}}(\bar{\boldsymbol{h}}_{G,i}, \bar{\boldsymbol{h}}_{G',c'(j)}) \right]$$

$$\overset{\text{Eq. (A.10)}}{=} \frac{1}{N!} \sum_{c' \in P_N} \left[ \frac{1}{N!} \frac{1}{N} \sum_{j=1}^{N} (N-1)! \sum_{i=1}^{N} k_{\text{NN}}(\bar{\boldsymbol{h}}_{G,i}, \bar{\boldsymbol{h}}_{G',c'j}) \right]$$

$$= \frac{1}{N!} \sum_{c' \in P_N} \left[ \frac{1}{N^2} \sum_{j=1}^{N} \sum_{i=1}^{N} k_{\text{NN}}(\bar{\boldsymbol{h}}_{G,i}, \bar{\boldsymbol{h}}_{G',c'(j)}) \right].$$

Now, we consider the restricted permutations $P_N | j \to i'$ and repeat a similar treatment for $c' \in P_N$,

$$\bar{k}(G, G') = \frac{1}{N^2} \sum_{j=1}^{N} \frac{1}{N!} \sum_{c' \in P_N} \sum_{i=1}^{N} k_{\text{NN}}(\bar{\boldsymbol{h}}_{G,i}, \bar{\boldsymbol{h}}_{G',c'(j)})$$

$$= \frac{1}{N^2} \sum_{j=1}^{N} \frac{1}{N!} \sum_{i'=1}^{N} \sum_{c' \in P_N | j \to i'} \sum_{i=1}^{N} k_{\text{NN}}(\bar{\boldsymbol{h}}_{G,i}, \bar{\boldsymbol{h}}_{G',c'(j)})$$

$$= \frac{1}{N^2} \sum_{j=1}^{N} \frac{1}{N!} \sum_{i'=1}^{N} \sum_{c' \in P_N | j \to i'} \sum_{i=1}^{N} k_{\text{NN}}(\bar{\boldsymbol{h}}_{G,i}, \bar{\boldsymbol{h}}_{G',i'})$$

$$= \frac{1}{N^2} \sum_{j=1}^{N} \frac{1}{N!} \sum_{i'=1}^{N} (N-1)! \sum_{i=1}^{N} k_{\text{NN}}(\bar{\boldsymbol{h}}_{G,i}, \bar{\boldsymbol{h}}_{G',i'})$$

$$= \frac{1}{N^2} \frac{1}{N!} \sum_{i'=1}^{N} N! \sum_{i=1}^{N} k_{\text{NN}}(\bar{\boldsymbol{h}}_{G,i}, \bar{\boldsymbol{h}}_{G',i'})$$

$$= \frac{1}{N^2} \sum_{i'=1}^{N} \sum_{i=1}^{N} k_{\text{NN}}(\bar{\boldsymbol{h}}_{G,i}, \bar{\boldsymbol{h}}_{G',i'}).$$

$\square$

**Lemma A.3** (Mercer Decomposition of the GNTK). *The GNTK is Mercer and can be decomposed as*

$$k_{\text{GNN}}(G, G') = \sum_{r=0}^{\infty} \lambda_r \sum_{s=1}^{M(d,r)} Z_{s,r}(\bar{\boldsymbol{h}}_G) Z_{s,r}(\bar{\boldsymbol{h}}_{G'})$$

*where $\lambda_k$ are identical to eigenvalues of $k_{\text{NN}}$. The algebraic multiplicity of each $\lambda_r$ is $M(d,r)$. The eigenfunctions $\{Z_{s,r}\}_{s \le M(d,r)}$ are degree-$r$ polynomials with the permutation invariant additive structure*

$$Z_{s,r}(\bar{\boldsymbol{h}}_G) := \frac{1}{N} \sum_{j=1}^{N} Y_{s,r}(\bar{\boldsymbol{h}}_{G,j}).$$

*where $Y_{s,r}$ are degree-$r$ spherical harmonics.*

***Proof of Lemma A.3.*** Plugging in the Mercer decomposition of $k_{\text{NN}}$ as given in Eq. (A.3) into Proposition 3.1 we get,

$$k_{\text{GNN}}(G, G') = \frac{1}{N^2} \sum_{j,j'=1}^{N} \sum_{r=0}^{\infty} \lambda_r \sum_{s=1}^{M(d,r)} Y_{s,r}(\bar{\boldsymbol{h}}_{G,j}) Y_{s,r}(\bar{\boldsymbol{h}}_{G',j'})$$

$$= \sum_{r=0}^{\infty} \lambda_r \sum_{s=1}^{M(d,r)} \left( \frac{1}{N} \sum_{j=1}^{N} Y_{s,r}(\bar{\boldsymbol{h}}_{G,j}) \right) \left( \frac{1}{N} \sum_{j'=1}^{N} Y_{s,r}(\bar{\boldsymbol{h}}_{G',j'}) \right) \quad \text{(A.11)}$$

$$= \sum_{r=0}^{\infty} \lambda_r \sum_{s=1}^{M(d,r)} Z_{s,r}(\bar{\boldsymbol{h}}_G) Z_{s,r}(\bar{\boldsymbol{h}}_{G'}).$$

$\square$

## A.2 Recursive Expression for the NTK

For the sake of completeness, we provide a closed-form expression for the NTK function used in Eq. (4) (for more details, see Section 2.1 in [5]). We limit the input space to $\mathbb{S}^{d-1}$ since, by the definition, our feature vectors are always normalized, i.e., $\|\bar{\boldsymbol{h}}_u\|_2 = 1$ for every $u \in V(G)$. For a ReLU network with $L$ layers considered in Eq. (A.1) with inputs on the sphere (by taking appropriate

limits on the widths), the corresponding $k_{\mathrm{NN}}(\boldsymbol{x}, \boldsymbol{x}')$ ([21]) depends on $\angle(\boldsymbol{x}, \boldsymbol{x}')$ and is given by $k_{\mathrm{NN}}(\boldsymbol{x}, \boldsymbol{x}') = \kappa_{\mathrm{NN}}^{(L)}(\boldsymbol{x}^T \boldsymbol{x}')$ where $\kappa_{\mathrm{NN}}^{(L)}(\cdot)$ is defined recursively as follows:

$$
\begin{aligned}
\kappa_{\mathrm{NN}}^{(1)}(u) &= \kappa^{(1)}(u) = u, \\
\kappa^{(l)}(u) &= \kappa_1\big(\kappa^{(l-1)}(u)\big), \\
\kappa_{\mathrm{NN}}^{(l)}(u) &= \kappa_{\mathrm{NN}}^{(l-1)}(u) \cdot \kappa_0\big(\kappa^{(l-1)}(u)\big) + \kappa^{(l)}(u) \quad \text{for } 2 \le l \le L,
\end{aligned}
\tag{A.12}
$$

where

$$
\begin{aligned}
\kappa_0(u) &= \frac{1}{\pi}\big(\pi - \arccos(u)\big), \\
\kappa_1(u) &= \frac{1}{\pi}\Big(u(\pi - \arccos(u)) + \sqrt{1 - u^2}\Big).
\end{aligned}
$$

Finally, we note that $\kappa_{\mathrm{NN}}^{(L)}(1) = 1$ (Bietti and Bach [5]), and hence $k_{\mathrm{NN}}(\boldsymbol{x}, \boldsymbol{x}') \le 1$ for all $\boldsymbol{x}, \boldsymbol{x}' \in \mathbb{S}^{d-1}$.

## A.3 Effect of Structure on the Hypothesis Space

In Section 4, we demonstrated that the additive permutation invariant structure of $k_{\mathrm{GNN}}$ help produce tighter bandit regret and information gain bounds, when the reward function is also permutation invariant. We now characterize how this invariance alters the hypothesis space, independent of the bandit setup. Bietti and Bach [5] give the Mercer decomposition of an NTK defined on a $Nd$-dimensional sphere. Applying their result we may decompose $\kappa_{\mathrm{NN}}$ as

$$
\kappa_{\mathrm{NN}}(G, G') = \sum_{r \ge 0}^{\infty} \lambda_{\mathrm{NN},r} \sum_{s=1}^{N(Nd,r)} Y_{s,r,Nd}(\bar{\boldsymbol{h}}_G) Y_{s,r,Nd}(\bar{\boldsymbol{h}}_{G'})
$$

where $Y_{s,r} : \mathbb{S}^{Nd-1} \to \mathbb{R}$ is the $s$-th degree-$r$ spherical harmonic polynomial. Each eigenvalue $\lambda_{\mathrm{NN},r}$ corresponds to the eigenspace $V_{Nd,r}$, the space of degree-$r$ spherical harmonics, defined on an $Nd$-dimensional domain. The algebraic multiplicity of each eigenvalue is $M(Nd, r)$ and equal to the dimension of its eigenspace,

$$
\dim(V_{Nd,r}) = M(Nd, r) = \frac{2k + d - 2}{r}\binom{r + d - 3}{d - 2} = cr^{Nd-2}.
$$

Lastly, the eigenvalues decay at a $\lambda_{\mathrm{NN},r} \simeq c(Nd, L)r^{-Nd}$ rate. The dependence of $c(Nd, L)$ on $Nd$ is exponential, but linear in $L$, as shown by [5].

We compare this kernel to the GNTK, which has the invariances encoded in its construction. In Lemma A.3 we show that it may be written as

$$
k_{\mathrm{GNN}}(G, G') = \sum_{r \ge 0}^{\infty} \lambda_{\mathrm{GNN},r} \sum_{s=1}^{N(d,r)} Z_{s,r,Nd}(\bar{\boldsymbol{h}}_G) Z_{s,r,Nd}(\bar{\boldsymbol{h}}_{G'}),
$$

where the eigenvalues $\lambda_{\mathrm{GNN},r}$ decay at a $c(d, L)r^{-d}$ rate, and have an algebraic multiplicity of $M(d, r) \simeq cr^{d-2}$. The eigenvectors $Z_{s,r}$ are degree-$r$ polynomials with the permutation invariant additive structure

$$
Z_{s,r,Nd}(\bar{\boldsymbol{h}}_G) := \frac{1}{N}\sum_{j=1}^{N} Y_{s,r,d}(\bar{\boldsymbol{h}}_{G,j}).
$$

where $Y_{s,r,d}$ are degree-$r$ spherical harmonics, defined on $\mathbb{S}^{d-1}$. Let $\bar{V}_{Nd,r}$ be the eigenspace corresponding to $\lambda_{\mathrm{GNN},r}$ the $r$-th eigenvalue. Due to the specific structure of the $Z_{s,r,Nd}$ polynomials, there exists a bijection between $\bar{V}_{Nd,r}$ and $V_{d,r}$ the space of degree-$r$ spherical harmonics defined on $\mathbb{S}^{d-1}$. The two vector spaces are isomorphic and thus have the same (finite) dimensionality, $\dim(\bar{V}_{Nd,r}) = \dim(V_{d,r})$. This implies that the $r$-th eigenspace of the GNTK is smaller than the $r$-th eigenspace of the NTK

$$
\frac{\dim(\bar{V}_{Nd,r})}{\dim(V_{Nd,r})} = \frac{\dim(V_{d,r})}{\dim(V_{Nd,r})} \le \frac{r^d}{r^{Nd}} = \frac{1}{r^{d(N-1)}}.
$$

Further, note that $V_{d,r} \subsetneq V_{Nd,r}$. This connection is shown in Figure 4.

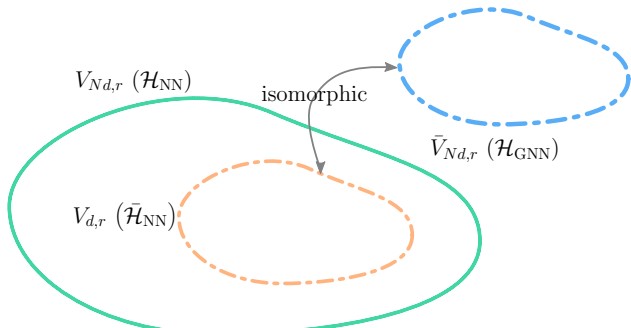

Figure 4: Relation between the GNN and NN hypothesis space

# B   Information Gain Bounds

In this section, we present a bound for the maximum information gain (defined in Eq. (5)) of the graph neural tangent kernel using the fact that a GNTK can be decomposed as an average of lower-dimensional NTKs (see Eq. (4)).

***Proof of Theorem 4.1***. We follow a similar technique as in Vakili et al. [41]. Consider an arbitrary sequence of graphs $(G_i)_{i \leq T}$, where each $G_i \in \mathcal{G}$. Let $\boldsymbol{K}_{\text{GNN}} \in \mathbb{R}^{T \times T}$ denote the corresponding GNTK matrix where

$$[\boldsymbol{K}_{\text{GNN}}]_{i,j} = k_{\text{GNN}}(G_i, G_j) = \frac{1}{N^2} \sum_{u,u'=1}^{N} \sum_{r=0}^{\infty} \lambda_r \sum_{s=1}^{M(d,r)} Y_{s,r}(\bar{\boldsymbol{h}}_{G_i,u}) Y_{s,r}(\bar{\boldsymbol{h}}_{G_j,u'}),$$

holds by Lemma A.3. We decompose $k_{\text{GNN}}(G_i, G_j)$ into $k_D(G_i, G_j) + k_O(G_i, G_j)$ where

$$k_D(G_i, G_j) = \frac{1}{N^2} \sum_{u,u'=1}^{N} \sum_{r \leq D} \lambda_r \sum_{s=1}^{M(d,r)} Y_{s,r}(\bar{\boldsymbol{h}}_{G_i,u}) Y_{s,r}(\bar{\boldsymbol{h}}_{G_j,u'}),$$

$$k_O(G_j, G_j) = \frac{1}{N^2} \sum_{u,u'=1}^{N} \sum_{r \geq D+1} \lambda_r \sum_{s=1}^{M(d,r)} Y_{s,r}(\bar{\boldsymbol{h}}_{G_i,u}) Y_{s,r}(\bar{\boldsymbol{h}}_{G_j,u'}).$$

The kernel $k_D$ is reproducing for $\mathcal{H}_{k_D}$, a finite-dimensional subspace of $\mathcal{H}_{k_{\text{GNN}}}$ that is spanned by the eigenfunctions corresponding to the first $D$ *distinct* eigenvalues. The kernel $K_O$ is reproducing for $\mathcal{H}_{k_O}$ which is orthogonal to $\mathcal{H}_{k_D}$. Moreover $k_D(G_i, G_j) = \phi_D^T(G_i) \phi_D(G_j)$ where the concatenated feature vector $\phi_D(G)$ is defined as

$$\phi_D(G) = \left( \left( \frac{\sqrt{\lambda_0}}{N} \sum_{u=1}^{N} Y_{s,0}(\bar{\boldsymbol{h}}_{G,u}), \right)_{s \leq N(d,0)}, \cdots, \left( \frac{\sqrt{\lambda_D}}{N} \sum_{u=1}^{N} Y_{s,D}(\bar{\boldsymbol{h}}_{G,u}) \right)_{s \leq N(d,D)} \right).$$

Here, $\phi_D(G) \in \mathbb{R}^{\tilde{D}}$ where

$$\tilde{D} = \sum_{r=0}^{D} M(d,r) \simeq C \sum_{r=0}^{D} r^{d-2} \leq C \frac{(D+1)^{d-1}}{D-1}, \tag{B.1}$$

since by Stirling's approximation $M(d,r)$ grows with $r^{d-2}$ [5].

Recall that the information gain is $I(\boldsymbol{y}_T; \boldsymbol{f}_T) = \frac{1}{2} \log \det(\boldsymbol{I} + \lambda^{-1} \boldsymbol{K}_{\text{GNN}})$. Defining $\boldsymbol{K}_D$ and $\boldsymbol{K}_O$ such that $[\boldsymbol{K}_D]_{i,j} = k_D(G_i, G_j)$ and $[\boldsymbol{K}_O]_{i,j} = k_O(G_i, G_j)$, we get $\boldsymbol{K}_{\text{GNN}} = \boldsymbol{K}_D + \boldsymbol{K}_O$ and therefore,

$$\begin{aligned} I(\boldsymbol{y}_T; \boldsymbol{f}_T) &= \frac{1}{2} \log \det(\boldsymbol{I} + \lambda^{-1}(\boldsymbol{K}_D + \boldsymbol{K}_O)) \\ &= \frac{1}{2} \log \det(\boldsymbol{I} + \lambda^{-1} \boldsymbol{K}_D) + \frac{1}{2} \log \det(\boldsymbol{I} + (\boldsymbol{I} + \lambda^{-1} \boldsymbol{K}_D)^{-1} \boldsymbol{K}_O). \end{aligned} \tag{B.2}$$

We bound each term separately, starting with the first term.

Consider the $T \times \tilde{D}$ feature matrix $\boldsymbol{\Phi}_D = [\boldsymbol{\phi}_D(G_1), \cdots, \boldsymbol{\phi}_D(G_T)]$. Then $\boldsymbol{K}_D = \boldsymbol{\Phi}_D \boldsymbol{\Phi}_D^T$ and by the Weinstein-Aronszajn identity,

$$\frac{1}{2} \log \det(\boldsymbol{I} + \lambda^{-1} \boldsymbol{K}_D) = \frac{1}{2} \log \det(\boldsymbol{I} + \lambda^{-1} \boldsymbol{\Phi}_D^T \boldsymbol{\Phi}_D).$$

For positive definite matrices $\boldsymbol{P} \in \mathbb{R}^{n \times n}$, we have $\log \det \boldsymbol{P} \leq n \log \operatorname{tr}(\boldsymbol{P}/n)$. Applying this identity we get,

$$
\begin{aligned}
\frac{1}{2} \log \det(\boldsymbol{I} + \lambda^{-1} \boldsymbol{K}_D) &\leq \frac{1}{2} \tilde{D} \log \left( 1 + \frac{\lambda^{-1}}{\tilde{D}} \operatorname{tr} \left( \boldsymbol{\Phi}_D^T \boldsymbol{\Phi}_D \right) \right) \\
&= \frac{1}{2} \tilde{D} \log \left( 1 + \frac{\lambda^{-1}}{\tilde{D}} \sum_{t=1}^{T} \boldsymbol{\phi}_D^T(G_t) \boldsymbol{\phi}_D(G_t) \right) \\
&= \frac{1}{2} \tilde{D} \log \left( 1 + \frac{\lambda^{-1}}{\tilde{D}} \sum_{t=1}^{T} \|\boldsymbol{\phi}_D(G_t)\|_2^2 \right) \\
&\leq \frac{1}{2} \tilde{D} \log \left( 1 + \frac{\lambda^{-1}}{\tilde{D}} \sum_{t=1}^{T} \sum_{r=0}^{D} \frac{\lambda_r}{N^2} \sum_{s=1}^{M(d,r)} \left( \sum_{u=1}^{N} Y_{s,r}(\bar{\boldsymbol{h}}_{G_t,u}) \right)^2 \right) \\
&\leq \frac{1}{2} \tilde{D} \log \left( 1 + \frac{\lambda^{-1}}{\tilde{D}} \sum_{t=1}^{T} \sum_{u,u'=1}^{N} \sum_{r=0}^{D} \frac{\lambda_r}{N^2} \sum_{s=1}^{M(d,r)} Y_{s,r}(\bar{\boldsymbol{h}}_{G_t,u}) Y_{s,r}(\bar{\boldsymbol{h}}_{G_t,u'}) \right) \\
&= \frac{1}{2} \tilde{D} \log \left( 1 + \frac{\lambda^{-1}}{\tilde{D}} \sum_{t=1}^{T} k_D(G_t, G_t) \right) \\
&\leq \frac{1}{2} \tilde{D} \log \left( 1 + \frac{T/\lambda}{\tilde{D}} \right).
\end{aligned}
$$
(B.3)

The last inequality holds since by definition, $k_{\text{GNN}}$ is uniformly bounded by 1 on the unit sphere (this holds since $k_{\text{NN}}$ is also uniformly bounded by 1 on the same domain; see Appendix A.2).

For bounding the second term, we again use the $\log \det \boldsymbol{P} \leq n \log \operatorname{tr}(\boldsymbol{P}/n)$ inequality and write,

$$
\begin{aligned}
\frac{1}{2} \log \det(\boldsymbol{I} + (\boldsymbol{I} + \lambda^{-1} \boldsymbol{K}_D)^{-1} \boldsymbol{K}_O) &\leq \frac{T}{2} \log \left( 1 + \frac{\operatorname{tr}\left( (\boldsymbol{I} + \lambda^{-1} \boldsymbol{K}_D)^{-1} \boldsymbol{K}_O \right)}{T} \right) \\
&\leq \frac{T}{2} \log \left( 1 + \operatorname{tr}\left( \boldsymbol{K}_O \right) / T \right).
\end{aligned}
$$

The second inequality holds due to $(\boldsymbol{I} + \lambda^{-1} \boldsymbol{K}_D)^{-1}$ being positive definite, with eigenvalues smaller than 1. To bound $\operatorname{tr}(\boldsymbol{K}_O)$, note that

$$
\begin{aligned}
[\boldsymbol{K}_O]_{i,j} &= \frac{1}{N^2} \sum_{u,u'=1}^{N} \sum_{r \geq D+1} \lambda_r \sum_{s=1}^{M(d,r)} Y_{s,r}(\bar{\boldsymbol{h}}_{G_i,u}) Y_{s,r}(\bar{\boldsymbol{h}}_{G_j,u'}) \\
&= \frac{1}{N^2} \sum_{u,u'=1}^{N} \sum_{r \geq D+1} \lambda_r M(d,r) P_r(\langle \bar{\boldsymbol{h}}_{G_i,u}, \bar{\boldsymbol{h}}_{G_j,u'} \rangle) \\
&\leq \frac{1}{N^2} \sum_{u,u'=1}^{N} \sum_{r \geq D+1} \lambda_r M(d,r) \\
&\leq \sum_{r \geq D+1} \lambda_r M(d,r)
\end{aligned}
$$

The second equality follows from $\sum_{s=1}^{M(d,r)} Y_{s,r}(\boldsymbol{x}) Y_{s,r}(\boldsymbol{x}') = M(d,r) P_r(\boldsymbol{x}^T \boldsymbol{x}')$, where $P_r$ is the degree-$r$ Legendre polynomial [5]. The Legendre basis is bounded in $[0, 1]$, resulting in the

first inequality. By Bietti and Bach [5, Corollary 3], there exists a constant $c_1(d, L)$ such that $\lambda_r \leq c_1(d, L)r^{-d}$. Stirling's approximation states that $n! \sim \sqrt{2\pi n}(n/e)^n$. Therefore, there exists a constant $c_2$ such that $M(d, r) \leq c_2 r^{d-2}$. Therefore, there exists a constant $c(d, L)$ such that,

$$[\boldsymbol{K}_O]_{i,j} \leq \sum_{r \geq D+1} \lambda_r M(d, r) \leq c(d, L) \sum_{r \geq D+1} r^{-2} \leq \frac{c(d, L)}{D}$$

where the second inequality comes from

$$\sum_{r \geq D+1} r^{-2} \leq \int_D^\infty z^{-2} \mathrm{d}z = \frac{1}{D}.$$

Therefore, we may bound the second term of the information gain as follows,

$$\frac{1}{2} \log \det(\boldsymbol{I} + (\boldsymbol{I} + \lambda^{-1}\boldsymbol{K}_D)^{-1}\boldsymbol{K}_O) \leq \frac{T}{2} \log \left(1 + \frac{c(d, L)}{D}\right) \tag{B.4}$$

From Eq. (B.2), Eq. (B.3) and Eq. (B.4),

$$I(\boldsymbol{y}_T; \boldsymbol{f}_T) \leq \frac{1}{2}\tilde{D} \log \left(1 + \frac{T\lambda^{-1}}{\tilde{D}}\right) + \frac{T}{2} \log \left(1 + \frac{c(d, L)}{D}\right). \tag{B.5}$$

For the first term to dominate the second, $\tilde{D}$ has to be set to

$$\tilde{D} = \left\lceil \left(\frac{c(d, L)T}{\log(1 + T/\lambda)}\right)^{\frac{d-1}{d}} \right\rceil.$$

This results in

$$\gamma_T = \mathcal{O}\left(\left(\frac{T}{\log(1 + \frac{T}{\lambda})}\right)^{\frac{d-1}{d}} \log \left(1 + \frac{T}{\lambda}\left(\frac{\log(1 + \frac{T}{\lambda})}{T}\right)^{\frac{d-1}{d}}\right)\right),$$

that is itself $\mathcal{O}(T^{\frac{d-1}{d}} \log^{\frac{1}{d}} T)$, therefore concluding the proof. $\qquad\square$

## C  Proof of Theorem 4.2 and Theorem 4.3

***Proof of Theorem 4.3***. The proposed algorithm GNN-PE (see Algorithm 1) is a variant of the Phased GP Uncertainty algorithm proposed in [7]. The following analysis closely follows the one of [7] (but ignores misspecification), with important differences pertained to the introduction of GNN estimator and GNTK analysis. The algorithm runs in episodes of exponentially increasing length $T_e$, and maintains a set of potentially optimal graphs $\mathcal{G}_e$. To compute the set of potentially optimal graphs after every episode, it uses the confidence bounds from Theorem 4.2. The total number of episodes is denoted with $E$, and it holds that $E \leq \lceil \log_2 T \rceil$, since the length of the episode is growing exponentially.

To bound the regret of GNN-PE, we make use of the finite-dimensional tangent kernel. With a slightly different notation from Section 3, we set $\hat{k}_{\mathrm{GNN}}(\cdot, \cdot) = \boldsymbol{g}_{\mathrm{GNN}}^T(\cdot; \boldsymbol{\theta}^0)\boldsymbol{g}_{\mathrm{GNN}}(\cdot; \boldsymbol{\theta}^0)/m$, where $\boldsymbol{g}_{\mathrm{GNN}}^T(\cdot; \boldsymbol{\theta}^0)$ denotes the gradient of the GNN at initialization. We argued in the main text that this kernel can well approximate $k_{\mathrm{GNN}}$. The feature map corresponding to this kernel, $\hat{\phi}(G) = \boldsymbol{g}_{\mathrm{GNN}}(G)/m$ can be viewed as a finite-dimensional approximation of $\phi_{\mathrm{GNN}}$, the (infinite length) feature map of the GNTK.

Throughout the proof, we denote the posterior mean and variance calculated via $\mathrm{GP}(0, \hat{k}_{\mathrm{GNN}})$ by $\hat{\mu}_{t-1}$ and $\hat{\sigma}_{t-1}$, respectively. Recall that the posterior mean and variance function after observing the data $(G_i, y_i)_{i<t}$ is calculated via

$$\begin{aligned}
\hat{\mu}_{t-1}(G; \hat{k}_{\mathrm{GNN}}) &= \hat{\boldsymbol{k}}_{t-1}^T(G)(\hat{\boldsymbol{K}}_{t-1} + \lambda\boldsymbol{I})^{-1}\boldsymbol{y}_{t-1}, \\
\hat{\sigma}_{t-1}^2(G; \hat{k}_{\mathrm{GNN}}) &= \hat{k}_{\mathrm{GNN}}(G, G) - \hat{\boldsymbol{k}}_{t-1}^T(G)(\hat{\boldsymbol{K}}_{t-1} + \lambda\boldsymbol{I})^{-1}\hat{\boldsymbol{k}}_{t-1}(G),
\end{aligned} \tag{C.4}$$

where the constant $\lambda$ is the variance proxy for observation noise. Here $\boldsymbol{y}_{t-1} = [y_i]_{i<t}$ is the vector of observed values, $\hat{\boldsymbol{k}}_{t-1}(\boldsymbol{x}) = [\hat{k}_{\mathrm{GNN}}(\boldsymbol{x}, \boldsymbol{x}_\tau)]_{i<t}$, and $\hat{\boldsymbol{K}}_{t-1} = [\hat{k}_{\mathrm{GNN}}(\boldsymbol{x}_i, \boldsymbol{x}_j)]_{i,j<t}$ is the kernel matrix.

---

**Algorithm 1:** GNN PHASED ELIMINATION (GNN-PE)

---

**Input:** $m$, $J$, $\eta$, $\lambda$, $T$

Set episode index $e = 1$, episode length $T_e = 1$, and set of potentially optimal graphs $\mathcal{G}_e = \mathcal{G}$

**Initialize** *network parameters to a random $\boldsymbol{\theta}^0$.*

**for** $t = 1, \ldots, T_e$ **do**

> For all $G \in \mathcal{G}$, calculate $\hat{\sigma}_{t-1}(G)$ as defined in Eq. (8).
> Select

$$G_t = \arg\max_{G \in \mathcal{G}_e} \hat{\sigma}^2_{t-1}(G) \tag{C.1}$$

**end**

Receive $\{y_1, \ldots, y_{T_e}\}$, such that

$$y_t = f^*(G_t) + \epsilon_t \quad \text{for} \quad t \in \{1, \ldots, T_e\}$$

Calculate $\boldsymbol{\theta}^{(J)}_e = \text{TrainGNN}\left(m, J, \eta, \lambda, \boldsymbol{\theta}^0, (G_t, y_t)^{T_e}_{t=1}\right)$

Use $\beta$ and $\varepsilon$ as defined in Equation (C.7) and Equation (C.8) and update

$$\mathcal{G}_{e+1} \leftarrow \left\{ G \in \mathcal{G}_e : f_{\text{GNN}}(G; \boldsymbol{\theta}^{(J)}_e) + \beta\hat{\sigma}_{T_e}(G) + 2\varepsilon \geq \max_{G \in \mathcal{G}_e} \left( f_{\text{GNN}}(G; \boldsymbol{\theta}^{(J)}_e) - \beta\hat{\sigma}_{T_e}(G) \right) \right\}, \tag{C.2}$$

Set $T_{e+1} \leftarrow 2T_e$, $e \leftarrow e + 1$ and return to the **Initialize** step

**Output:** Terminate after $T$ total evaluations ($E$ total episodes) and return

$$\hat{G}_T = \arg\max_{G \in \mathcal{G}} f_{\text{GNN}}(G; \boldsymbol{\theta}^{(J)}_{E-1}) \tag{C.3}$$

---

---

**Algorithm 2:** TrainGNN

---

**Input:** $m$, $J$, $\eta$, $\lambda$, $\boldsymbol{\theta}^0$, $(G_i, y_i)_{i<t}$

Define $\mathcal{L}(\boldsymbol{\theta}) = \frac{1}{t}\sum_{i<t}\left(f_{\text{GNN}}(G_i; \boldsymbol{\theta}) - y_i\right)^2 + m\lambda\|\boldsymbol{\theta} - \boldsymbol{\theta}^0\|^2_2$

Initialize $\boldsymbol{\theta}^{(0)} = \boldsymbol{\theta}^0$

**for** $j = 1, \ldots, J$ **do**

> $\boldsymbol{\theta}^{(j)} = \boldsymbol{\theta}^{(j-1)} - \eta\nabla\mathcal{L}(\boldsymbol{\theta}^{(j-1)})$

**end**

**Output:** $\boldsymbol{\theta}^{(J)}$

---

We note that GNN-PE uses $\hat{\sigma}_{t-1}$ as the variance estimate, while instead of $\hat{\mu}_{t-1}$, the algorithm makes use of the GNN predictions, i.e., $f_{\text{GNN}}(G; \boldsymbol{\theta}^{(J)}_t)$ for the center of the confidence set.

As will become clear soon, since GNN-PE uses $\hat{\sigma}_{t-1}$, this yields a regret bound depending on $\hat{\gamma}_T$, the information gain corresponding to the approximate kernel $\hat{k}_{\text{GNN}}$. Lastly, in Lemma C.1, for appropriately set width $m$, we bound $\hat{\gamma}_T$ with $\gamma_{\text{GNN},T}$ (maximum information gain corresponding to the exact GNTK). We use this result in our steps bellow.

**Step 1 (Max variance bound)** Consider any fixed episode $e$, and recall that $T_e$ denotes the episode length. By the exploration policy of GNN-PE (Equation (C.1)), at any step $t$ (within an episode) and for any graph $G \in \mathcal{G}_e$, we have $\hat{\sigma}_{t-1}(G) \leq \hat{\sigma}_{t-1}(G_t)$. From Eq. (C.4), since the covariance matrix is positive definite, conditioning on a larger set of points reduces the posterior variance and thus $\hat{\sigma}_{T_e}(G) \leq \hat{\sigma}_{t-1}(G)$, for all $G \in \mathcal{G}_e$ and $t \leq T_e$. Putting the two inequalities together, $\hat{\sigma}_{T_e}(G) \leq \min_{t \leq T_e} \hat{\sigma}_{t-1}(G_t)$, which gives

$$\hat{\sigma}^2_{T_e}(G) \leq \frac{1}{T_e}\sum_{t=1}^{T_e}\hat{\sigma}^2_{t-1}(G_t).$$

For any $s \in [0, 1/\lambda]$, it holds that

$$s^2 \leq \frac{1}{\lambda\log(1 + 1/\lambda)}\log(1 + s^2).$$

For any $G_t$, we have $\hat{\sigma}^2_{t-1}(G_t)/\lambda \leq \hat{k}_{\text{GNN}}(G_t, G_t)/\lambda \leq 1/\lambda$. Therefore,

$$\hat{\sigma}^2_{t-1}(G_t) \leq \frac{1}{\log(1 + 1/\lambda)} \log(1 + \lambda^{-2}\hat{\sigma}^2_{t-1}(G_t)).$$

From Srinivas et al. [37, Lemma 5.3], we can conclude that,

$$\sum_{t=1}^{T_e} \log(1 + \lambda^{-1}\hat{\sigma}^2_{t-1}(G_t)) = 2I(G_1, \cdots, G_t; \hat{k}_{\text{GNN}}) \leq 2\hat{\gamma}_{T_e},$$

where $\hat{\gamma}_{T_e}$ is the maximum information gain corresponding to this episode and the kernel $\hat{k}_{\text{GNN}}$. This inequality allows us to bound the posterior variance at the end of episode $e$ as follows. For any $G \in \mathcal{G}_e$,

$$\hat{\sigma}^2_{T_e}(G) \leq \frac{2\hat{\gamma}_{T_e}}{T_e \log(1 + \lambda^{-1})}. \tag{C.5}$$

**Step 2 (Confidence bounds)** Consider an episode $e$, and let $\tilde{\delta} = \delta/(3E)$, where $E \leq \log T$ is the number of episode. From Theorem C.2, for all $G \in \mathcal{G}$, with probability at least $1 - \tilde{\delta}$,

$$|f_{\text{GNN}}(G; \boldsymbol{\theta}_e^{(J)}) - f^*(G)| \leq \beta\hat{\sigma}_{T_e}(G) + \varepsilon, \tag{C.6}$$

where for simplicity we use

$$\beta := \sqrt{2}B + \frac{\sigma}{\sqrt{\lambda}}\sqrt{2\log 2|\mathcal{G}|/\tilde{\delta}} + \sqrt{\frac{2B}{m\eta\lambda}}\left(\sqrt{2} + (1 - m\eta\lambda)^{J/2}\right), \tag{C.7}$$

$$\varepsilon := \tilde{C}L^3\left(\frac{B}{m\lambda}\right)^{2/3}\sqrt{m\log m}. \tag{C.8}$$

Applying the union bound Eq. (C.6) holds for every $G \in \mathcal{G}$ and $e \in [E]$ with probability at least $1 - \delta/3$. Finally, by using $\tilde{\delta} = \delta/3$ in Lemma C.1 and by applying the union bound, we have that both events in Lemma C.1 and in Eq. (C.6) hold jointly with probability at least $1 - 2\delta/3$. In the rest of the proof, we condition on the joint event holding true. This implies that $G^* \in \mathcal{G}_e$ for every $e \in [E]$, i.e., according to the rule in Eq. (C.2), the algorithm will not eliminate $G^*$.

**Step 3 (Cumulative Regret)** We use $R_e$ to denote the episodic regret, and write

$$R_T = \sum_{e=1}^{E} R_e \leq m_1 B + \sum_{e=2}^{E}\sum_{t=1}^{T_e}\left(f^*(G^*) - f^*(G_t^{(e)})\right). \tag{C.9}$$

The first inequality follows since $k_{\text{GNN}}$ is uniformly bounded by 1 and the RKHS norm of $f^*$ is bounded by $B$. We also add an additional superscript $e$ in $G_t^{(e)}$, to denote a graph selected in episode $e$ at time step $t$.

Consider any episode $e$, the following holds due to Eq. (C.6):

$$f^*(G^*) - f^*(G_t^{(e)}) \leq f_{\text{GNN}}(G^*; \boldsymbol{\theta}_{e-1}^{(J)}) + \beta\hat{\sigma}_{T_{e-1}}(G^*) + 2\varepsilon$$
$$- \left(f_{\text{GNN}}(G_t^{(e)}; \boldsymbol{\theta}_{e-1}^{(J)}) - \beta\hat{\sigma}_{T_{e-1}}(G_t^{(e)})\right).$$

Moreover, we use the elimination rule in Eq. (C.2) to obtain:

$$f^*(G^*) - f^*(G_t^{(e)}) \leq \left(f_{\text{GNN}}(G^*; \boldsymbol{\theta}_{e-1}^{(J)}) - \beta\hat{\sigma}_{T_{e-1}}(G^*) - \varepsilon\right)$$
$$+ 2\beta\hat{\sigma}_{T_{e-1}}(G^*) + 4\varepsilon$$
$$- \left(f_{\text{GNN}}(G_t^{(e)}; \boldsymbol{\theta}_{e-1}^{(J)}) + \beta\hat{\sigma}_{T_{e-1}}(G_t^{(e)}) + \varepsilon\right)$$
$$+ 2\beta\hat{\sigma}_{T_{e-1}}(G_t^{(e)})$$
$$\leq 4\beta_{T_{e-1}+1}\max_{G \in \mathcal{G}_{e-1}}\hat{\sigma}_{T_{e-1}}(G) + 4\varepsilon. \tag{C.10}$$

Next, we combine Eq. (C.10) and Eq. (C.9) and write:

$$R_T \leq m_1 B + \sum_{e=2}^{E} \sum_{t=1}^{T_e} 4\beta_{T_{e-1}+1} \max_{G \in \mathcal{G}_{e-1}} \hat{\sigma}_{T_{e-1}}(G) + 4\varepsilon \tag{C.11}$$

$$\leq m_1 B + \sum_{e=2}^{E} 4T_e \beta \sqrt{\frac{2\hat{\gamma}_{T_{e-1}}}{T_{e-1} \log(1+\lambda^{-1})}} + 4T_e\varepsilon \tag{C.12}$$

$$= m_1 B + \sum_{e=2}^{E} 8\beta \sqrt{\frac{2T_{e-1}\hat{\gamma}_{T_{e-1}}}{\log(1+\lambda^{-1})}} + 4T_e\varepsilon \tag{C.13}$$

$$\leq m_1 B + \sum_{e=2}^{E} 8\beta \sqrt{\frac{2T\hat{\gamma}_T}{\log(1+\lambda^{-1})}} + 4T_e\varepsilon \tag{C.14}$$

$$\leq m_1 B + 8(\log(T)+1)\left( \beta \sqrt{\frac{2T\hat{\gamma}_T}{\log(1+\lambda^{-1})}} + 4T\varepsilon \right) \tag{C.15}$$

$$\leq m_1 B + 8(\log(T)+1)\left( \beta \sqrt{\frac{2T(\gamma_T+\epsilon(m))}{\log(1+\lambda^{-1})}} + 4T\varepsilon \right), \tag{C.16}$$

where Eq. (C.12) follows since $T > m_e$ for every $e$ and Eq. (C.5) and Eq. (C.13) since $T_e = 2T_{e-1}$. To obtain Eq. (C.14), we use that $T > T_{e-1}$ and $\hat{\gamma}_T \geq \hat{\gamma}_{T_e-1}$. Finally, Eq. (C.15) follows since the number of episodes $E \leq \lceil \log T \rceil$, and Eq. (C.16) follows from Lemma C.1.

**Step 4 (Putting everything together)**  Plugging in the expressions for $\beta$ and $e$ we obtain

$$R_T \leq 8\sqrt{2}(\log T+1)\sqrt{\frac{2T(\gamma_T+\epsilon(m))}{\log(1+\lambda^{-1})}} \left( B + \frac{\sigma}{\sqrt{\lambda}} \sqrt{\log \frac{|\mathcal{G}|(\log T+1)}{\delta}} \sqrt{\frac{B}{m\eta\lambda}} \left( \sqrt{2} + (1-m\eta\lambda)^{J/2} \right) \right)$$

$$+ B + 32(\log T+1)\tilde{C}TL^3 \left( \frac{B}{m\lambda} \right)^{2/3} \sqrt{m \log m}$$

Since $m = \text{poly}(t)$ and $\eta \sim 1/m$, the last term and $\epsilon(m)$ vanishes with $T$ at a $o(1)$ rate, and the gradient descent error term becomes a constant factor. Then we obtain with probability at least $1 - \delta$

$$R_T = \mathcal{O}\left( \sqrt{T\gamma_T} \left( B + \frac{\sigma}{\sqrt{\lambda}} \sqrt{\log \frac{|\mathcal{G}| \log T}{\delta}} \right) \right). \tag{C.17}$$

$\square$

**Lemma C.1 (Bounding MIG with its approximation).** *Set $\delta \in (0,1)$. If $m = \text{poly}\left(t, L, |\mathcal{G}|, \lambda, B, \lambda_0^{-1}, \log(N/\delta)\right)$, then with probability at least $1 - \delta$,*

$$\hat{\gamma}_T \leq \gamma_T + \epsilon(m),$$

*where $\gamma_T$ is the maximum information gain of the GNTK over $\mathcal{G}$ as defined in Eq. (5), and $\epsilon(m) = o(m^{-1/4})$.*

***Proof of Lemma C.1.*** The proof follows from Lemma C.6, by repeating the technique given in Lemma D.5, [25]. Here we repeat it for the sake of completeness.

Consider an arbitrary sequence of graphs $(G_t)_{t \leq T}$. Consider the feature map $\hat{\phi}(G) = g_{\text{GNN}}(G; \theta^0)/\sqrt{m}$. For the kernel $\hat{k}_{\text{GNN}}$, which corresponds to this feature map, the information gain after observing $T$ samples is

$$\hat{I}_T = \frac{1}{2} \log \det \left( \mathbf{I} + \lambda^{-1} \bar{\mathbf{G}}_T \bar{\mathbf{G}}_T^T / m \right),$$

where $\bar{G}_T = [\bar{g}(G_t)]_{t \leq T}^T$. Let $[K_{\text{GNN}}]_{i,j \leq T} = k_{\text{GNN}}(G_i, G_j)$ with $k$ the NTK function of the fully-connected $L$-layer network.

$$
\begin{aligned}
\hat{I}_T &= \frac{1}{2} \log \det \left( I + \lambda^{-1} K_{\text{GNN}} + \lambda^{-1}(\bar{G}_T \bar{G}_T^T/m - K_{\text{GNN}}) \right) \\
&\overset{(a)}{\leq} \frac{1}{2} \log \det \left( I + \lambda^{-1} K_{\text{GNN}} \right) + \langle (I + \lambda^{-1} K_{\text{GNN}})^{-1}, \lambda^{-1}(\bar{G}_T \bar{G}_T^T/m - K_{\text{GNN}}) \rangle \\
&\leq I_T + \lambda^{-1} \left\| (I + \lambda^{-1} K_{\text{GNN}})^{-1} \right\|_F \left\| \bar{G}_T \bar{G}_T^T/m - K_{\text{GNN}} \right\|_F \\
&\overset{(b)}{\leq} I_T + \lambda^{-1} \sqrt{T} \left\| \bar{G}_T \bar{G}_T^T/m - K_{\text{GNN}} \right\|_F \\
&\overset{(c)}{\leq} I_T + \lambda^{-1} T \sqrt{T} \epsilon \\
&\overset{(d)}{\leq} \gamma_T + \epsilon(m).
\end{aligned}
\tag{C.18}
$$

Inequality (a) holds by concavity of $\log \det(\cdot)$. Inequality (b) holds since $I \preccurlyeq I + \lambda^{-1} K_{\text{GNN}}$. Inequality (c) holds due to Lemma C.6. Finally, inequality (d) uses the polynomial choice of $m$, and requires that $m$ grows with at least $O(T^6)$. Equation C.18 holds for any arbitrary context set, thus it also holds for the sequence which maximizes the information gain. $\square$

## C.1   Proof of Theorem 4.2

We first present the formal version of Theorem 4.2.

**Theorem C.2** (GNN Confidence Bound, Formal). *Set $\delta \in (0,1)$. Suppose $f^* \in \mathcal{H}_{k_{\text{GNN}}}$ with a bounded norm $\|f^*\|_{k_{\text{GNN}}} \leq B$. Samples of $f$ are observed with zero-mean $\sigma^2$-sub-Gaussian noise. Assume that the random sequences $(G_i)_{i<t}$ and $(\epsilon_i)_{i<t}$ are statistically independent. Set $J > 1$, choose the width $m = \text{poly}\left(t, L, |\mathcal{G}|, \lambda, \lambda_0^{-1}, \log(N/\delta)\right)$, and learning rate $\eta = C(Lm + m\lambda)^{-1}$ with some universal constant $C$. Then for all graphs $G \in \mathcal{G}$, with probability of at least $1 - \delta$,*

$$
|f^*(G) - \hat{\mu}_{t-1}(G)| \leq \beta \hat{\sigma}_t(G) + \tilde{C} L^3 \left( \frac{B}{m\lambda} \right)^{2/3} \sqrt{m \log m}
$$

*where*

$$
\beta = \sqrt{2}B + \frac{\sigma}{\sqrt{\lambda}} \sqrt{2 \log\left(2|\mathcal{G}|/\delta\right)} + \sqrt{\frac{2B}{m\eta\lambda}} \left( \sqrt{2} + (1 - m\eta\lambda)^{J/2} \right)
$$

*for some constant $\bar{C}$.*

To prove the theorem, first we state the necessary lemmas.

**Lemma C.3** (**Confidence interval for $f_{\text{GNN}}$ around $\hat{\mu}_{t-1}$**). *Assume history $H_t = \{(G_i, y_i)\}_{i \leq t}$ with $(G_i)_{i \leq t}$ and $(\varepsilon_i)_{i \leq t}$ statistically independent. Let $m = \text{poly}\left(t, L, \lambda, \log(|\mathcal{G}|N/\delta)\right)$. There exists $C_1$, such that for any $\delta > 0$, if the learning rate is picked $\eta = C_1(Lm + m\lambda)^{-1}$, then for a graph $G \in \mathcal{G}$, with probability of at least $1 - \delta$,*

$$
|f_{\text{GNN}}(G; \boldsymbol{\theta}^{(J)}) - \hat{\mu}_t(G)| \leq \hat{\sigma}_t(G) \sqrt{\frac{2B}{m\eta\lambda}} \left( \sqrt{2} + (1 - m\eta\lambda)^{J/2} \right) + \tilde{C} L^3 \left( \frac{B}{m\lambda} \right)^{2/3} \sqrt{m \log m}
$$

*for some constant $\bar{C}$, where $\hat{\mu}_t$ and $\hat{\sigma}_t$ are as defined in Eq. (C.4).*

***Proof of Lemma C.3.*** We define $Z := \lambda I + \bar{G}^T \bar{G}/(Tm)$ and $b = \sum_{i \leq t} y_i \bar{g}(G_i)/(T\sqrt{m})$. Recall that $\bar{G}_T = [\bar{g}(G_t)]_{t \leq T}^T$. Let the sequence $(\tilde{\boldsymbol{\theta}}^{(j)})_{j=1}^J$ denote the gradient descent updates on the following loss function

$$
\tilde{\mathcal{L}}(\boldsymbol{\theta}) = \frac{1}{t} \sum_{i \leq t} \left( \langle \bar{g}(G_i), \boldsymbol{\theta} - \boldsymbol{\theta}^0 \rangle - y_i \right)_2^2 + m\lambda \left\| \boldsymbol{\theta} - \boldsymbol{\theta}^0 \right\|_2^2.
$$

Note that $\tilde{\boldsymbol{\theta}}^0 = \boldsymbol{\theta}^0$ and $\tilde{\boldsymbol{\theta}}^{(j)}$ also depends on $t$ the number of data points. We omit the $t$ index, for simplicity of the notation during the proof of this lemma.

By Lemma C.7, $\left\|\bar{\boldsymbol{G}}\right\|_F \leq C\sqrt{TLm}$ and we have,

$$\boldsymbol{Z} \stackrel{\text{w.h.p}}{\preccurlyeq} (\lambda + CL)\boldsymbol{I} \preccurlyeq \frac{1}{m\eta}\boldsymbol{I}, \tag{C.19}$$

since $\eta$ is set such that $\eta \leq C(m\lambda + Lm)^{-1}$. Therefore, for any $\boldsymbol{x} \in \mathbb{R}^p$, $\|\boldsymbol{x}\|_{\boldsymbol{Z}} \leq \frac{1}{\sqrt{m\eta}}\|\boldsymbol{x}\|_2$. Now consider $\boldsymbol{x}, \boldsymbol{x}' \in \mathbb{R}^p$, using Eq. (C.19) together with Cauchy-Schwarz implies,

$$\langle \boldsymbol{x}, \boldsymbol{x}'\rangle \leq \|\boldsymbol{x}\|_{\boldsymbol{Z}}\|\boldsymbol{x}'\|_{\boldsymbol{Z}^{-1}} \stackrel{\text{w.h.p}}{\leq} \frac{1}{\sqrt{mn}}\|\boldsymbol{x}\|_2\|\boldsymbol{x}'\|_{\boldsymbol{Z}^{-1}}. \tag{C.20}$$

Applying the inequality above, we may write

$$\begin{aligned}
\langle \bar{\boldsymbol{g}}(G), \boldsymbol{\theta}^{(J)} - \boldsymbol{\theta}^0\rangle &= \langle \bar{\boldsymbol{g}}(G), \boldsymbol{\theta}^{(J)} - \tilde{\boldsymbol{\theta}}^{(J)}\rangle + \langle \bar{\boldsymbol{g}}(G), \tilde{\boldsymbol{\theta}}^{(J)} - \boldsymbol{\theta}^0\rangle \\
&\stackrel{\text{w.h.p}}{\leq} \frac{1}{\sqrt{m\eta}}\|\bar{\boldsymbol{g}}(G)\|_{\boldsymbol{Z}^{-1}}\left\|\boldsymbol{\theta}^{(J)} - \tilde{\boldsymbol{\theta}}^{(J)}\right\|_2 + \langle \bar{\boldsymbol{g}}(G), \tilde{\boldsymbol{\theta}}^{(J)} - \boldsymbol{\theta}^0\rangle \\
&\stackrel{\text{w.h.p}}{\leq} 2\left\|\frac{\bar{\boldsymbol{g}}(G)}{\sqrt{m}}\right\|_{\boldsymbol{Z}^{-1}}\sqrt{\frac{B}{m\eta\lambda}} + \langle \bar{\boldsymbol{g}}(G), \tilde{\boldsymbol{\theta}}^{(J)} - \boldsymbol{\theta}^0\rangle
\end{aligned} \tag{C.21}$$

For the last inequality of Eq. (C.21) we have used Lemma C.8. Decomposing the second term of the right hand side in Eq. (C.21) gives,

$$\begin{aligned}
\langle \bar{\boldsymbol{g}}(G), \tilde{\boldsymbol{\theta}}^{(J)} - \boldsymbol{\theta}^0\rangle &= \langle \bar{\boldsymbol{g}}(G), \frac{\boldsymbol{Zb}}{\sqrt{m}}\rangle + \langle \bar{\boldsymbol{g}}(G), \tilde{\boldsymbol{\theta}}^{(J)} - \boldsymbol{\theta}^0 - \frac{\boldsymbol{Zb}}{\sqrt{m}}\rangle \\
&\stackrel{\text{w.h.p}}{\leq} \frac{\bar{\boldsymbol{g}}^T(G)\boldsymbol{Zb}}{\sqrt{m}} + \frac{1}{\sqrt{\eta}}\left\|\frac{\bar{\boldsymbol{g}}(G)}{\sqrt{m}}\right\|_{\boldsymbol{Z}^{-1}}\left\|\tilde{\boldsymbol{\theta}}^{(J)} - \boldsymbol{\theta}^0 - \frac{\boldsymbol{Zb}}{\sqrt{m}}\right\|_2 \\
&\stackrel{\text{w.h.p}}{\leq} \frac{\bar{\boldsymbol{g}}^T(G)\boldsymbol{Zb}}{\sqrt{m}} + \left\|\frac{\bar{\boldsymbol{g}}(G)}{\sqrt{m}}\right\|_{\boldsymbol{Z}^{-1}}\sqrt{\frac{2B}{m\eta\lambda}}(1 - \eta m\lambda)^{J/2}
\end{aligned} \tag{C.22}$$

where the first inequlity is a consequence of Eq. (C.20). The second inequality follows from the convergence of GD on the proxy loss $\tilde{\mathcal{L}}$, given in Lemma C.8. By the definition of posterior mean and variance (Eq. C.4) when the regularization parameter is set to $\lambda \leftarrow t\lambda$ we have,

$$\hat{\mu}_t(G) = \frac{\bar{\boldsymbol{g}}^T(G)\boldsymbol{Zb}}{\sqrt{m}},$$

$$\hat{\sigma}_t(G) = \left\|\frac{\bar{\boldsymbol{g}}(G)}{\sqrt{m}}\right\|_{\boldsymbol{Z}^{-1}}.$$

The final upper bound on $f_{\text{GNN}}(G; \boldsymbol{\theta}^{(J)}) - \hat{\mu}_t(G)$ follows from plugging in Equation C.22 into Equation C.21, and applying Lemma C.10. Similarly, for the lower bound we have,

$$-f_{\text{GNN}}(G; \boldsymbol{\theta}^{(J)}) \stackrel{\text{w.h.p}}{\leq} \langle \bar{\boldsymbol{g}}(G), \boldsymbol{\theta}^0 - \boldsymbol{\theta}^{(J)}\rangle + \tilde{C}L^3\left(\frac{B}{m\lambda}\right)^{2/3}\sqrt{m\log m} \tag{C.23}$$

$$\langle \bar{\boldsymbol{g}}(G), \boldsymbol{\theta}^0 - \tilde{\boldsymbol{\theta}}^{(J)}\rangle \stackrel{\text{w.h.p}}{\leq} -\hat{\mu}_t(G) + \hat{\sigma}_t(G)\sqrt{\frac{2B}{m\eta\lambda}}(1 - \eta m\lambda)^{J/2} \tag{C.24}$$

$$\langle \bar{\boldsymbol{g}}(G), \boldsymbol{\theta}^0 - \boldsymbol{\theta}^{(J)}\rangle \stackrel{\text{w.h.p}}{\leq} 2\hat{\sigma}_t(G)\sqrt{\frac{B}{m\eta\lambda}} + \langle \bar{\boldsymbol{g}}(G), \tilde{\boldsymbol{\theta}}^{(J)} - \boldsymbol{\theta}^0\rangle \tag{C.25}$$

Where inequality C.23 holds by Lemma C.7, and the next two inequalities are driven similarly to equations C.21 and C.22. The lower bound results by putting together equations C.23-C.25, and this concludes the proof. Note that we are implicitly taking a union bound over the 6 inequalities that all hold with high probability. The conditions of the used lemmas require that $m$ is picked at a $\text{poly}(\log(N/\delta))$ rate, and a constant number of union bounds do not affect this rate. $\qquad\square$

The next lemma gives a confidence interval over members of $\mathcal{H}_{k_{\text{GNN}}}$.

**Lemma C.4** (RKHS Confidence Interval from Vakili et al. [40]). *Let $f^* \in \mathcal{H}_k$ with $\|f^*\|_k \leq B$, and the observation noise to be sub-gaussian with parameter $\sigma^2$. Assume $H_t = \{(\boldsymbol{x}_\tau, y_\tau)\}$ with $(\boldsymbol{x}_\tau)_{\tau \leq t}$ and $(\varepsilon_\tau)_{\tau \leq t}$ statistically independent. Then for a fixed input $\boldsymbol{x}$, with probability greater than $1 - \delta$,*

$$|f^*(\boldsymbol{x}) - \mu_t(\boldsymbol{x})| \leq (B + \frac{\sigma}{\sqrt{\lambda}}\sqrt{2\log(2/\delta)})\sigma_t(\boldsymbol{x}).$$

The following lemma shows that members of $\mathcal{H}_{k_{\text{GNN}}}$ are well described by the first-order Taylor approximation of a GNN around initialization.

**Lemma C.5** (**Approximation by a linearized GNN**). *Let $f$ be a member of $\mathcal{H}_{k_{\text{GNN}}}$ with bounded RKHS norm $\|f\|_{k_{\text{GNN}}} \leq B$. Set $\delta \in (0,1)$ and let $N$ denote an upper bound on the possible number of nodes for a graph. If $m = \mathcal{O}\left(L^6|\mathcal{G}|^4/\lambda_0^4 \log(|\mathcal{G}|^2 LN/\delta)\right)$, then with probability greater than $1 - \delta$, there exists $\boldsymbol{\theta}^* \in \mathbb{R}^p$ such that for all $G \in \mathcal{G}$*

$$f(G) = \langle \boldsymbol{g}_{\text{GNN}}(G; \boldsymbol{\theta}^0), \boldsymbol{\theta}^* \rangle, \quad \sqrt{m}\|\boldsymbol{\theta}^*\|_2 \leq \sqrt{2}B.$$

***Proof of Lemma C.5.*** The proof follows the technique for Lemma 5.1 Zhou et al. [45] with some modifications.

From Eq. (C.30) proof of Lemma C.6, for $m = \mathcal{O}(L^6/\epsilon^4 \log(LN/\delta))$, and for $G_i$ and $G_j$ in the domain,

$$\left|k_{\text{GNN}}(G_i, G_j) - \boldsymbol{g}_{\text{GNN}}^T(G_i; \boldsymbol{\theta}^0)\boldsymbol{g}_{\text{GNN}}(G_j; \boldsymbol{\theta}^0)\right| \leq (L+1)\epsilon$$

with probability greater than $1 - \delta$. Let $\boldsymbol{K}_{\text{full}}$ be the GNTK matrix calculated for all $G \in \mathcal{G}$ and $\bar{\boldsymbol{G}}_{\text{full}} = [\boldsymbol{g}_{\text{GNN}}^T(G; \boldsymbol{\theta}^0)]_{G \in \mathcal{G}}$. Then applying a union bound over all $G \in \mathcal{G}$, and setting $\delta \leftarrow \delta/|\mathcal{G}|^2$, if $m = \mathcal{O}(L^6/\epsilon^4 \log(|G|^2 LN/\delta))$, then

$$\left\|\boldsymbol{K}_{\text{full}} - \bar{\boldsymbol{G}}_{\text{full}}^T\bar{\boldsymbol{G}}_{\text{full}}/m\right\|_F \leq |\mathcal{G}|\epsilon$$

with probability greater than $1 - \delta$. Now applying this inequality when $\epsilon = \lambda_0/2|\mathcal{G}|$, we get that if

$$m = \mathcal{O}(L^6|\mathcal{G}|^4/\lambda_0^4 \log(|G|^2 LN/\delta))$$

then $\left\|\boldsymbol{K}_{\text{full}} - \bar{\boldsymbol{G}}_{\text{full}}^T\bar{\boldsymbol{G}}_{\text{full}}/m\right\|_F \leq \lambda_0/2$, with probability greater than $1 - \delta$. Via the Triangle inequality we get that

$$\bar{\boldsymbol{G}}_{\text{full}}^T\bar{\boldsymbol{G}}_{\text{full}}/m \succcurlyeq \boldsymbol{K}_{\text{full}} - \left\|\boldsymbol{K}_{\text{full}} - \bar{\boldsymbol{G}}_{\text{full}}^T\bar{\boldsymbol{G}}_{\text{full}}/m\right\|_F \succcurlyeq \boldsymbol{K}_{\text{full}} - \frac{\lambda_0}{2} \succcurlyeq \boldsymbol{K}_{\text{full}}/2 \succ 0. \quad \text{(C.26)}$$

Since $\lambda_0 > 0$, $\bar{\boldsymbol{G}}_{\text{full}}$ is positive definite and may be decomposed as $\bar{\boldsymbol{G}}_{\text{full}} = \boldsymbol{PAQ}^T$, where $\boldsymbol{P} \in \mathbb{R}^{p \times |\mathcal{G}|}$, $\boldsymbol{P} \in \mathbb{R}^{|\mathcal{G}| \times |\mathcal{G}|}$ are unitary and $\boldsymbol{A} \succ 0$. Let $\boldsymbol{f} = [f(G)]_{G \in \mathcal{G}}$ be the vector of function values. We show that $\boldsymbol{\theta}^* = \boldsymbol{PA}^{-1}\boldsymbol{Q}^T\boldsymbol{f}$ satisfies the statement of the lemma. By definition of $\boldsymbol{\theta}^*$,

$$\bar{\boldsymbol{G}}_{\text{full}}^T\boldsymbol{\theta}^* = \boldsymbol{QAP}^T\boldsymbol{PA}^{-1}\boldsymbol{Q}^T\boldsymbol{f} = \boldsymbol{f}$$

which implies for all $G \in \mathcal{G}$, $\langle \boldsymbol{g}_{\text{GNN}}(G; \boldsymbol{\theta}^0), \boldsymbol{\theta}^* \rangle = f_{\text{GNN}}(G)$. As for the norm of $\boldsymbol{\theta}^*$ we may write,

$$\|\boldsymbol{\theta}^*\|_2^2 \leq \boldsymbol{f}^T\boldsymbol{QA}^{-2}\boldsymbol{Q}^T = \boldsymbol{f}^T(\bar{\boldsymbol{G}}_{\text{full}}^T\bar{\boldsymbol{G}}_{\text{full}}^T)^{-1}\boldsymbol{f} \leq \frac{2}{m}\boldsymbol{f}^T\boldsymbol{K}_{\text{full}}^{-1}\boldsymbol{f} \leq \frac{2B^2}{m}$$

where the next to last inequality holds due to Eq. (C.26), and the last inequality follows from $\|f\|_{k_{\text{GNN}}}^2 \leq B^2$. □

We are now ready to present the proof of our main confidence interval bound.

***Proof of Theorem 4.2.*** Consider Lemma C.4, when the kernel function is $\tilde{k}(G, G') = \bar{g}^T(G)\bar{g}(G')/m$ and choose the regularization parameter $t\lambda$. Subsequently, the posterior mean and variance after observing $t$ samples will be $\hat{\mu}_t(G)$ and $\hat{\sigma}_t(G)$. Then this lemma states that for $f \in \mathcal{H}_{\tilde{k}}$ with a norm bounded by $B$, with probability greater than $1 - \delta/2$,

$$|f(G) - \tilde{\mu}_t(G)| \leq \hat{\sigma}_t\left(B + \frac{\sigma}{\sqrt{\lambda}}\sqrt{2\log 4/\delta}\right).$$

By Lemma C.5, for $m$ large enough the reward function can be written as

$$f^*(G) = \langle \boldsymbol{g}_{\text{GNN}}(G; \boldsymbol{\theta}^0), \boldsymbol{\theta}^* \rangle, \quad \sqrt{m} \|\boldsymbol{\theta}^*\|_2 \leq \sqrt{2}B.$$

for all $G \in \mathcal{G}$, indicating that $f^* \in \mathcal{H}_{\hat{k}_{\text{GNN}}}$ with $\|f\|_{\hat{k}_{\text{GNN}}} \leq \sqrt{2}B$. Therefore, following Lemma C.4 with probability greater than $1 - \delta/2$,

$$|f^*(G) - \hat{\mu}_t(G)| \leq \hat{\sigma}_t \left( \sqrt{2}B + \frac{\sigma}{\sqrt{\lambda}} \sqrt{2\log 4/\delta} \right)$$

for some fixed $G$. Further, Lemma C.3 bounds the difference between $\hat{\mu}_{t-1}(\cdot)$ and $\hat{\mu}_{t-1}(\cdot)$ with probability higher than $1 - \delta/2$. Plugging in $\hat{\mu}_{t-1}$ and $\hat{\sigma}_{t-1}$ gives,

$$|\hat{\mu}_{t-1}(G) - f^*(G)| \leq \hat{\sigma}_t(G) \left[ \sqrt{2}B + \frac{\sigma}{\lambda} \sqrt{2\log 4/\delta} + \sqrt{\frac{2B}{m\eta\lambda}} \left( \sqrt{2} + (1 - m\eta\lambda)^{J/2} \right) \right]$$
$$+ \tilde{C}L^3 \left( \frac{B}{m\lambda} \right)^{2/3} \sqrt{m \log m}$$

with probability greater than $1 - \delta$. Setting $\delta \leftarrow \delta/|\mathcal{G}|$ and taking a union bound over all $G \in \mathcal{G}$ concludes the proof of Theorem C.2. The informal version of the theorem is achieved by considering that $m = \text{poly}(t)$ and omitting all the terms that are $o(t^{-1})$ with $t$. $\qquad\square$

## C.2 GNN Helper Lemmas

**Lemma C.6 (Norm concentration of Gram matrix and GNTK matrix at initialization).** *Set $\epsilon > 0$, $\delta \in (0, 1)$, and let $N = |V(G)|$ denote the number of nodes for every graph $G \in \mathcal{G}$. For width $m = \Omega(L^6/\epsilon^4 \log(LNT^2/\delta))$ in Eq. (2), the following holds with probability at least $1 - \delta$:*

$$\left\| \boldsymbol{K}_{\text{GNN}} - \bar{\boldsymbol{G}}^T \bar{\boldsymbol{G}}/m \right\|_F \leq T\epsilon.$$

***Proof of Lemma C.6.*** We make use of the connection between the GNN as defined in Eq. (2) and a fully-connected neural network. In particular, let $f_{\text{NN}}(\boldsymbol{x}; \boldsymbol{\theta}) : \mathbb{R}^d \to \mathbb{R}$ be a fully-connected network, with $L$ hidden layers of equal width $m$, and ReLU activations as defined in Eq. (A.1). Then, we have

$$f_{\text{GNN}}(G; \boldsymbol{\theta}) = \frac{1}{\sqrt{N}} \sum_{v \in V(G)} f_{\text{NN}}(\bar{\boldsymbol{h}}_v; \boldsymbol{\theta}).$$

Let $\bar{\boldsymbol{G}}$ be a matrix with $T$ columns where each column $i \in [T]$ contains gradient vector $\boldsymbol{g}_{\text{GNN}}(G_i; \boldsymbol{\theta}^0)$. Then, the matrix $\bar{\boldsymbol{G}}^T \bar{\boldsymbol{G}}$ is a $T \times T$ matrix and represents the Gram matrix of the network for the parameters $\boldsymbol{\theta}^0$. For all $i, j \leq T$, we have

$$[\bar{\boldsymbol{G}}^T \bar{\boldsymbol{G}}]_{i,j} = \frac{1}{N} \sum_{v=1}^{N} \boldsymbol{g}^T(\bar{\boldsymbol{h}}_v) \boldsymbol{g}(\bar{\boldsymbol{h}}_{v'}), \tag{C.27}$$

where $\boldsymbol{g}(\boldsymbol{x}) = \nabla_{\boldsymbol{\theta}} f_{\text{NN}}(\boldsymbol{x}; \boldsymbol{\theta}^0)$ denotes the gradient of $f_{\text{NN}}$ at initialization. It follows by the definition of the neural tangent kernel that

$$k_{\text{GNN}}(G_i, G_j) = \frac{1}{N} \sum_{v=1}^{N} k_{\text{NN}}(\bar{\boldsymbol{h}}_v, \bar{\boldsymbol{h}}_{v'}). \tag{C.28}$$

For some fixed $\epsilon > 0$ and $\delta \in (0, 1)$, the result of Arora et al. [2, Theorem 3.1] states that when $m = \Omega(L^6/\epsilon^4 \log(L/\delta))$, the following holds for any $\boldsymbol{x}, \boldsymbol{x}'$ with unit norms and probability at least $1 - \delta$:

$$\left| k_{\text{NN}}(\boldsymbol{x}, \boldsymbol{x}') - \boldsymbol{g}^T(\boldsymbol{x}) \boldsymbol{g}(\boldsymbol{x}')/m \right| \leq (L + 1)\epsilon. \tag{C.29}$$

Next, by using equations C.28 and C.27, and the triangle inequality for any two input graphs $G_i$, $G_j$ we get

$$\left| k_{\text{GNN}}(G_i, G_j) - \boldsymbol{g}_{\text{GNN}}^T(G_i; \boldsymbol{\theta}^0) \boldsymbol{g}_{\text{GNN}}(G_j; \boldsymbol{\theta}^0) \right| \leq \frac{1}{N} \sum_{v=1}^{N} \left| k_{\text{NN}}(\bar{\boldsymbol{h}}_v, \bar{\boldsymbol{h}}_{v'}) - \boldsymbol{g}^T(\bar{\boldsymbol{h}}_v) \boldsymbol{g}(\bar{\boldsymbol{h}}_{v'}) \right|.$$
$$\tag{C.30}$$

Then, since $\bar{h}_v \in \mathbb{S}^{d-1}$ for every node $v$ irrespective of the corresponding graph, we can use Eq. (C.29) together with the union bound over all $(\bar{h}_{G,u}, \bar{h}_{G',u})$ pairs and $m = \Omega(L^6/\epsilon^4 \log(LN/\delta))$, to obtain that for any $G_i$ and $G_j$ with probability at least $1 - \delta$:

$$\left| k_{\text{GNN}}(G_i, G_j) - g_{\text{GNN}}^T(G_i; \theta^0) g_{\text{GNN}}(G_j; \theta^0)/m \right| \leq (L+1)\epsilon.$$

To arrive at the main result we consider the difference in the Frobenius norm:

$$\left\| K_{\text{GNN}} - \bar{G}^T \bar{G}/m \right\|_F = \sqrt{\sum_{i,j \leq T} \left( k_{\text{GNN}}(G_i, G_j) - g_{\text{GNN}}^T(G_i; \theta^0) g_{\text{GNN}}(G_j; \theta^0)/m \right)^2}.$$

By setting $\epsilon \leftarrow \epsilon/(L+1)$ and again applying the union bound over each $(G_i, G_j)$ pair, for $m = \Omega(L^6/\epsilon^4 \log(LNT^2/\delta))$, the following holds with probability at least $1 - \delta$:

$$\left\| K_{\text{GNN}} - \bar{G}^T \bar{G}/m \right\|_F \leq T\epsilon.$$

$\square$

**Lemma C.7** (**Gradient descent norm bounds**). *Consider the fixed set $\{G_i\}_{i \leq t}$ of inputs. Let $\bar{G} = [g_{\text{GNN}}^T(G_i; \theta^0)]_{i \leq t}^T$ be the matrix of gradients at initialization and $\bar{G}^{(j)} = [g_{\text{GNN}}^T(G_i; \theta^{(j)})]_{i \leq t}^T$. The vector of network outputs after the $j$-th update is denoted by $f_{\text{GNN}}^{(j)} = [f_{\text{GNN}}(G_i; \theta^{(j)})]_{i \leq t}$. Assume $\tau$ is set such that $||\theta^{(j)} - \theta^0||_2 \leq \tau$ for all $j \leq J$. If $m = \text{poly}\left(t, L, \lambda^{-1}, \log(N/\delta)\right)$, then with probability greater than $1 - \delta$,*

$$\left\| \bar{G} \right\|_F \leq C_1 \sqrt{tmL} \tag{C.31}$$

$$\left\| \bar{G} - \bar{G}^{(j)} \right\|_F \leq C_2 \tau^{1/3} L^{7/2} \sqrt{tm \log m} \tag{C.32}$$

$$\left\| f_{\text{GNN}}^{(j)} - \bar{G}^{(j)}(\theta^{(j)} - \theta^0) \right\|_2 \leq C_3 \tau^{4/3} L^3 \sqrt{tm \log m} \tag{C.33}$$

*for some constants $C_1, C_2, C_3$.*

***Proof of Lemma C.7.*** We follow the recipe introduced in Zhou et al. [45], and reproduce gradient norm bounds for when the network is a GNN.

From Lemma B.3 Cao and Gu [10], we get $\left\| g_{\text{GNN}}(G_i; \theta^0) \right\|_2 \leq \bar{C}\sqrt{mL}$ with probability of at least $1 - \delta/3$. By the definition of Frobenius norm, it follows,

$$\left\| \bar{G} \right\|_F \leq \sqrt{t} \max_{i \leq t} C \left\| g_{\text{GNN}}(G_i; \theta^0) \right\|_2 \overset{\text{w.h.p}}{\leq} C\sqrt{tmL}.$$

For Eq. (C.32) we may write,

$$\left\| \bar{G}^{(j)} - \bar{G} \right\|_F \leq \sqrt{t} \max_{i \leq t} \left\| g_{\text{GNN}}(G_i; \theta^{(j)}) - g_{\text{GNN}}(G_i; \theta^0) \right\|_2$$

$$\leq \frac{\sqrt{t}}{N} \max_{i \leq t} \sum_{u \in V(G_i)} \left\| g(\bar{h}_v; \theta^{(j)}) - g(\bar{h}_v; \theta^0) \right\|_2$$

$$\overset{\text{w.h.p}}{\leq} \frac{\sqrt{t}}{N} \max_{i \leq t} \sum_{u \in V(G_i)} \tilde{C}_2 \tau^{1/3} L^3 \sqrt{\log m} \left\| g(\bar{h}_v; \theta^0) \right\|_2$$

$$\overset{\text{w.h.p}}{\leq} C_2 \tau^{1/3} L^{7/2} \sqrt{tm \log m}$$

with probability greater than $1 - \delta/3$, where the next to last inequality holds by Lemma B.5 Zhou et al. [45] and the last inequality follows directly from Lemma B.6 Zhou et al. [45].

As for Eq. (C.33), by definition of the single-BLOCK GNN and Lemma B.4 Zhou et al. [45],

$$\left\| f_{\text{GNN}}^{(j)} - \bar{G}^{(j)}(\theta^{(j)} - \theta^0) \right\|_2 \leq \sqrt{t} \max_{i \leq t} \left| f_{\text{GNN}}(G_i; \theta^{(j)}) - \langle g_{\text{GNN}}(G_i; \theta^{(j)}), \theta^{(j)} - \theta^0 \rangle \right|$$

$$\leq \frac{\sqrt{t}}{N} \max_{i \leq t} \sum_{u \in V(G_i)} \left| f(\bar{h}_v; \theta^{(j)}) - \langle g(\bar{h}_v; \theta^{(j)}), \theta^{(j)} - \theta^0 \rangle \right|$$

$$\overset{\text{w.h.p}}{\leq} C_3 \tau^{4/3} L^3 \sqrt{tm \log m}$$

with probability greater than $1 - \delta/3$. $\square$

**Lemma C.8** (**Convergence properties of the proxy optimization problem**). *Let the sequence* $(\tilde{\boldsymbol{\theta}}^{(j)})$ *denote the gradient descent updates on the following loss function,*

$$\tilde{\mathcal{L}}(\boldsymbol{\theta}) = \frac{1}{t} \sum_{i \leq t} \left( \langle \boldsymbol{g}_{\text{GNN}}(G_i; \boldsymbol{\theta}^0), \boldsymbol{\theta} - \boldsymbol{\theta}^0 \rangle - y_i \right)_2^2 + m\lambda \| \boldsymbol{\theta} - \boldsymbol{\theta}^0 \|_2^2.$$

*then if* $m = \text{poly}(T, L, B, |\mathcal{G}|, \lambda_0^{-1}, \lambda^{-1}, \log(N/\delta))$ *and the learning rate* $\eta \leq C(mL + m\lambda)^{-1}$

$$\left\| \tilde{\boldsymbol{\theta}}^{(j)} - \boldsymbol{\theta}^0 \right\|_2 \leq \sqrt{B/m\lambda},$$

$$\left\| \tilde{\boldsymbol{\theta}}^{(j)} - \boldsymbol{\theta}^0 - \left( \lambda \boldsymbol{I} + \bar{\boldsymbol{G}}^T \bar{\boldsymbol{G}}/(tm) \right)^{-1} \bar{\boldsymbol{G}}^T \boldsymbol{y}/(tm) \right\|_2 \leq (1 - \eta m\lambda)^{j/2} \sqrt{2B/m\lambda}.$$

*Proof of Lemma C.8.* This lemma adapts Lemma D.8 Kassraie and Krause [25] to our setting. We repeat the proof for the sake of completeness. Note that $\tilde{\mathcal{L}}$ is $m\lambda$-strongly convex, and $C(mL + m\lambda)$-smooth, since

$$\nabla^2 \tilde{\mathcal{L}} = \frac{2\bar{\boldsymbol{G}}^T \bar{\boldsymbol{G}}}{t} + 2m\lambda \boldsymbol{I} \leq 2\left( \frac{\|\bar{\boldsymbol{G}}\|_F^2}{t} + m\lambda \right) \boldsymbol{I} \leq C(mL + m\lambda)\boldsymbol{I},$$

where the second inequality follows from Lemma C.7. Strong Convexity of $\tilde{\mathcal{L}}$ guarantees a monotonic decrease of the loss if the learning rate is smaller than the smoothness coefficient inversed. Therefore,

$$\begin{aligned} m\lambda \left\| \tilde{\boldsymbol{\theta}}^{(j)} - \tilde{\boldsymbol{\theta}}^0 \right\|_2^2 &\leq m\lambda \left\| \tilde{\boldsymbol{\theta}}^{(j)} - \tilde{\boldsymbol{\theta}}^0 \right\|_2^2 + \frac{1}{t} \left\| \bar{\boldsymbol{G}}(\tilde{\boldsymbol{\theta}}^{(j)} - \tilde{\boldsymbol{\theta}}^0) - \boldsymbol{y} \right\|_2^2 \\ &\leq m\lambda \left\| \tilde{\boldsymbol{\theta}}^0 - \tilde{\boldsymbol{\theta}}^0 \right\|_2^2 + \frac{1}{t} \left\| \bar{\boldsymbol{G}}(\tilde{\boldsymbol{\theta}}^0 - \tilde{\boldsymbol{\theta}}^0) - \boldsymbol{y} \right\|_2^2 \\ &\leq \frac{\|\boldsymbol{y}\|_2^2}{t} \\ &\leq B \end{aligned}$$

From the RKHS assumption, the true reward is bounded by $B$ and hence the last inequality follows since the size of the training set is $t$.

Gradient descent on smooth and strongly convex functions converges to optima if the learning rate is smaller than the smoothness coefficient inversed. Under this condition the minima of $\tilde{\mathcal{L}}$ is unique and has the closed form

$$\tilde{\boldsymbol{\theta}}^* = \boldsymbol{\theta}^0 + \left( \lambda \boldsymbol{I} + \bar{\boldsymbol{G}}^T \bar{\boldsymbol{G}}/(mt) \right)^{-1} \bar{\boldsymbol{G}}^T \boldsymbol{y}/(mt)$$

Having set $\eta \leq C(mL + m\lambda)^{-1}$, we get that $\tilde{\boldsymbol{\theta}}^{(j)}$ converges to $\tilde{\boldsymbol{\theta}}^*$ with the following exponential rate,

$$\begin{aligned} \left\| \tilde{\boldsymbol{\theta}}^{(j)} - \boldsymbol{\theta}^0 - \left( \lambda \boldsymbol{I} + \bar{\boldsymbol{G}}^T \bar{\boldsymbol{G}}/(mt) \right)^{-1} \bar{\boldsymbol{G}}^T \boldsymbol{y}/(mt) \right\|_2^2 &\leq (1 - \eta m\lambda)^{(j)} \frac{2}{m\lambda} \left( \tilde{\mathcal{L}}(\boldsymbol{\theta}^0) - \tilde{\mathcal{L}}(\tilde{\boldsymbol{\theta}}^*) \right) \\ &\leq \frac{2(1 - \eta m\lambda)^j}{m\lambda} \frac{\|y\|_2^2}{t} \\ &\leq \frac{2B(1 - \eta m\lambda)^j}{m\lambda}. \end{aligned}$$

$\square$

**Lemma C.9** (**Gradient descent parameters bound**). *Let the sequence* $\boldsymbol{\theta}^{(J)}$ *denote the $J$-th gradient descent update on the GNN loss,*

$$\tilde{\mathcal{L}}(\boldsymbol{\theta}) = \frac{1}{t} \sum_{i \leq t} \left( f_{\text{GNN}}(G_i; \boldsymbol{\theta}^0) - y_i \right)_2^2 + m\lambda \| \boldsymbol{\theta} - \boldsymbol{\theta}^0 \|_2^2.$$

*If* $m = \text{poly}(T, L, B, |\mathcal{G}|, \lambda_0^{-1}, \lambda^{-1}, \log(N/\delta))$ *and* $\eta \leq C(mL + m\lambda)^{-1}$ *for some $C$, then with probability greater than* $1 - \delta$,

$$\left\| \boldsymbol{\theta}^{(J)} - \boldsymbol{\theta}^0 \right\|_2 \leq 2\sqrt{B/m\lambda}.$$

***Proof of Lemma C.9.*** Following Zhou et al. [45] we introduce the sequence $(\tilde{\boldsymbol{\theta}}^{(j)})$ which denotes the gradient descent updates on the following proxy loss,

$$\tilde{\mathcal{L}}(\boldsymbol{\theta}) = \frac{1}{t}\sum_{i \le t}\left(\langle \boldsymbol{g}_{\text{GNN}}(G_i;\boldsymbol{\theta}^0), \boldsymbol{\theta} - \boldsymbol{\theta}^0\rangle - y_i\right)_2^2 + m\lambda\|\boldsymbol{\theta} - \boldsymbol{\theta}^0\|_2^2.$$

By Lemma C.8,

$$\left\|\tilde{\boldsymbol{\theta}}^{(J)} - \boldsymbol{\theta}^0\right\|_2 \le \sqrt{\frac{B}{m\lambda}}$$

It remains to show that $\|\boldsymbol{\theta}^{(J)} - \tilde{\boldsymbol{\theta}}^{(J)}\|_2 \le \sqrt{B/m\lambda}$, which concludes the proof due to triangle inequality. By writing out the gradient descent updates of the two sequences we get,

$$
\begin{aligned}
\left\|\boldsymbol{\theta}^{(j+1)} - \tilde{\boldsymbol{\theta}}^{(j+1)}\right\|_2 &= \left\|(1 - \eta m\lambda)(\boldsymbol{\theta}^{(j)} - \tilde{\boldsymbol{\theta}}^{(j)}) - \frac{\eta}{t}(\bar{\boldsymbol{G}}^{(j)} - \bar{\boldsymbol{G}})^T(\boldsymbol{f}_{\text{GNN}}^{(j)} - \boldsymbol{y})\right. \\
&\quad \left. - \frac{\eta}{t}\bar{\boldsymbol{G}}^T\left(\boldsymbol{f}_{\text{GNN}}^{(j)} - \bar{\boldsymbol{G}}(\boldsymbol{\theta}^{(j)} - \boldsymbol{\theta}^0) + \bar{\boldsymbol{G}}(\boldsymbol{\theta}^{(j)} - \tilde{\boldsymbol{\theta}}^{(j)})\right)\right\|_2 \\
&\le \frac{\eta}{t}\left\|(\bar{\boldsymbol{G}}^{(j)} - \bar{\boldsymbol{G}})\right\|_2\left\|(\boldsymbol{f}_{\text{GNN}}^{(j)} - \boldsymbol{y})\right\|_2 + \frac{\eta}{t}\|\bar{\boldsymbol{G}}\|_2\left\|\boldsymbol{f}_{\text{GNN}}^{(j)} - \bar{\boldsymbol{G}}(\boldsymbol{\theta}^{(j)} - \boldsymbol{\theta}^0)\right\|_2 \\
&\quad + \left\|\boldsymbol{I} - \eta\left(m\lambda\boldsymbol{I} + \bar{\boldsymbol{G}}\bar{\boldsymbol{G}}^T/t\right)\right\|_2\left\|\boldsymbol{\theta}^{(j)} - \tilde{\boldsymbol{\theta}}^{(j)}\right\|_2
\end{aligned}
$$

We bound each term separately. In the rest of the proof, Lemma C.7 is always used with $\tau = \sqrt{B/m\lambda}$. Lemma C.3 Zhou et al. [45] directly holds for $f_{\text{GNN}}$, and states that $\|\boldsymbol{f}_{\text{GNN}}^{(j)} - \boldsymbol{y}\|_2 \le (B+1)\sqrt{t}$. Then Eq. (C.32) Lemma C.7 gives,

$$\frac{\eta}{t}\left\|(\bar{\boldsymbol{G}}^{(j)} - \bar{\boldsymbol{G}})\right\|_2\left\|(\boldsymbol{f}_{\text{GNN}}^{(j)} - \boldsymbol{y})\right\|_2 \overset{\text{w.h.p}}{\le} \eta C_1\left(\frac{B}{m\lambda}\right)^{1/6}L^{7/2}(B+1)\sqrt{m\log m}.$$

Recall that $f_{\text{GNN}}(G;\boldsymbol{\theta}^0) = 0$ by design. Then for the second term, using Equations C.31 and C.33 from Lemma C.7,

$$\frac{\eta}{t}\|\bar{\boldsymbol{G}}\|_2\left\|\boldsymbol{f}_{\text{GNN}}^{(j)} - \bar{\boldsymbol{G}}(\boldsymbol{\theta}^{(j)} - \boldsymbol{\theta}^0)\right\|_2 \overset{\text{w.h.p}}{\le} \eta C_2\left(\frac{B}{m\lambda}\right)^{2/3}L^{7/2}m\sqrt{\log m}.$$

As for the last term, first note that by Eq. (C.31) Lemma C.7, with high probability

$$\eta\left(m\lambda\boldsymbol{I} + \bar{\boldsymbol{G}}\bar{\boldsymbol{G}}^T/t\right) \overset{\text{w.h.p}}{\preccurlyeq} \eta(m\lambda\boldsymbol{I} + C_1 mL\boldsymbol{I}) \preccurlyeq \boldsymbol{I}$$

where the last inequality holds since $\eta$ is chosen to be small enough. Therefore,

$$\left\|\boldsymbol{I} - \eta\left(m\lambda\boldsymbol{I} + \bar{\boldsymbol{G}}\bar{\boldsymbol{G}}^T\right)\right\|_2\left\|\boldsymbol{\theta}^{(j)} - \tilde{\boldsymbol{\theta}}^{(j)}\right\|_2 \overset{\text{w.h.p}}{\le} (1 - \eta m\lambda)\left\|\boldsymbol{\theta}^{(j)} - \tilde{\boldsymbol{\theta}}^{(j)}\right\|_2.$$

We put the three terms back together and unroll the recursive inequality. Then if $m$ is picked to be large enough at the above stated polynomial rate,

$$\left\|\boldsymbol{\theta}^{(j)} - \tilde{\boldsymbol{\theta}}^{(j)}\right\|_2 \overset{\text{w.h.p}}{\le} \sqrt{\frac{B}{m\lambda}}.$$

$\square$

The next lemma shows that the first order approximation of a GNN at initialization can still describe the network after it has been trained with gradient descent for $J$ steps.

**Lemma C.10 (Taylor approximation of a trained GNN).** *If* $m = \text{poly}(T, L, B, |\mathcal{G}|, \lambda_0^{-1}, \lambda^{-1}, \log(N/\delta))$ *and for some constant* $C$, $\eta = C(mL + m\lambda)^{-1}$, *then*

$$\left|f_{\text{GNN}}(G_t;\boldsymbol{\theta}^{(J)}) - f_{\text{GNN}}(G_t;\boldsymbol{\theta}^0) - \langle \boldsymbol{g}_{\text{GNN}}(G_t;\boldsymbol{\theta}^0), \boldsymbol{\theta}^{(J)} - \boldsymbol{\theta}^0\rangle\right| \le \tilde{C}L^3\left(\frac{B}{m\lambda}\right)^{2/3}\sqrt{m\log m}$$

*with some constant* $\tilde{C}$ *and any* $t \le T$, *with probability greater than* $1 - \delta$.

***Proof of lemma C.10.*** By Lemma 4.1 Cao and Gu [10], if $m = \text{poly}(T, L, B, |\mathcal{G}|, \lambda_0^{-1}, \lambda^{-1}, \log(1/\delta))$ and $\eta$ is set according to the statement of the lemma, then for a fixed $\boldsymbol{x}$ with probability greater than $1 - \delta$,

$$\left| f_{\text{NN}}(\boldsymbol{x}; \boldsymbol{\theta}^{(J)}) - f_{\text{NN}}(\boldsymbol{x}; \boldsymbol{\theta}^0) - \langle g(\boldsymbol{x}; \boldsymbol{\theta}^0), \boldsymbol{\theta}^{(J)} - \boldsymbol{\theta}^0 \rangle \right| \leq C\tau^{4/3} L^3 \sqrt{m \log m}$$

where $\left\| \boldsymbol{\theta}^{(J)} - \boldsymbol{\theta}^0 \right\|_2 \leq \tau$. We use this inequality with $\boldsymbol{x} = (\bar{\boldsymbol{h}}_v^{(i)})$ for all $u \in V(G_i)$. Setting $\delta \leftarrow \delta/2N$ and applying the union bound gives

$$\left| \sum_{u \in V(G_i)} f_{\text{NN}}(\bar{\boldsymbol{h}}_v^{(i)}; \boldsymbol{\theta}^{(J)}) - \sum_{u \in V(G_i)} f_{\text{NN}}(\bar{\boldsymbol{h}}_v^{(i)}; \boldsymbol{\theta}^0) - \langle \sum_{u \in V(G_i)} \boldsymbol{g}(\bar{\boldsymbol{h}}_v^{(i)}; \boldsymbol{\theta}^0), \boldsymbol{\theta}^{(J)} - \boldsymbol{\theta}^0 \rangle \right|$$
$$\leq CN\tau^{4/3} L^3 \sqrt{m \log m}$$

Therefore, if $m = \text{poly}(T, L, B, |\mathcal{G}|, \lambda_0^{-1}, \lambda^{-1}, \log(N/\delta)))$ with probability greater than $1 - \delta$,

$$\left| f(G_i; \boldsymbol{\theta}^{(J)}) - f(G_i; \boldsymbol{\theta}^0) - \langle \boldsymbol{g}_{\text{GNN}}(G_i; \boldsymbol{\theta}^0), \boldsymbol{\theta}^{(J)} - \boldsymbol{\theta}^0 \rangle \right| \leq C\tau^{4/3} L^3 \sqrt{m \log m} \qquad \text{(C.34)}$$

It remains to bound $\|\boldsymbol{\theta}^{(J)} - \boldsymbol{\theta}^0\|_2$ in order to calculate $\eta$ in Equation C.34. From Lemma C.9, we have $\|\boldsymbol{\theta}^{(J)} - \boldsymbol{\theta}^0\|_2 \leq 2\sqrt{B/m\lambda}$. Setting $\delta \leftarrow \delta/T$ and taking a union bound over all $t \leq T$ concludes the proof. Note that the added $\log T$ term from union bound does not change rate for $m$, since it is already growing polynomially with $T$. $\qquad \square$

## D  Experiments

We include the details of the experiments in Section 5, together with the supplementary plots.

### D.1  Synthetic Permutation Invariant Datasets

To test our permutation invariant additive model, we pick the GNTK as the kernel function and create 18 datasets that inherit this structure. As explained in Section 5, each dataset consists of a finite domain of size $\mathcal{G} = 10000$ together with a reward function, both of which are generated randomly. The domains are sets of Erdős-Rényi random graphs, where each graph has $N$ nodes, and between each two nodes there exists an edge with probability $p$. The node features are i.i.d. $d$-dimensional standard Gaussian vectors.

For every domain, we sample a random reward function. We use $\text{GP}(0, k_{\text{GNN}})$ as a prior, and sample $f$ from its posterior GP. The posterior is calculated using a small random dataset $(G_i, y_i)_{i \leq 5}$, where $y_i$ are drawn independently from $\mathcal{N}(0, 1)$ and $G_i$ are randomly chosen from $\mathcal{G}_{p,N}$. We choose the posterior GP over the prior as it produces somewhat smoother samples. We note that functions drawn from this GP *do not* reside in $\mathcal{H}_{\text{GNN}}$. Table 2 shows the characteristics of the datasets, which will be released together with the code to generate them from scratch.

|           | $N = 5$ | $N = 20$ | $N = 100$ |
|-----------|---------|----------|-----------|
| $p = 0.05$ | $d \in \{10, 100\}$ | $d \in \{10, 100\}$ | $d \in \{10\}$ |
| $p = 0.2$  | $d \in \{10, 100\}$ | $d \in \{10, 100\}$ | $d \in \{10, 100\}$ |
| $p = 0.95$ | $d \in \{10, 100\}$ | $d \in \{10, 100\}$ | $d \in \{10, 100\}$ |

Table 2: Parameters of the synthetic datasets

### D.2  Practical Details

The python code to our algorithms, bandit environment, and experiments will be released.

**Algorithm.** There are some differences between how we utilize the algorithm in practice and the pseudo-code inAlgorithm 1. We list these modifications for transparency.

- When calculating $\hat{\sigma}_{t-1}$ we approximate $\hat{\boldsymbol{K}}_{t-1}$ with its diagonal so that the matrix inversion takes $o(t)$ operations.

- GNN-PE suggests to discard data from previous episodes, so that the decisions are non-adaptive. In practice we keep the history for training the network.

- We set all $T_e = 1$.

- Only from $t \geq T_2 = 80$ we follow Eq. (C.2) and intersect the sets of plausible maximizers. For the first $T_2$ steps construct them via

$$\mathcal{G}_{e+1} \leftarrow \left\{ G \in \mathcal{G} : f_{\mathrm{GNN}}(G; \boldsymbol{\theta}_e^{(J)}) + \beta_{T_e} \hat{\sigma}_{T_e}(G) \geq \max_{G \in \mathcal{G}} \left( f_{\mathrm{GNN}}(G; \boldsymbol{\theta}_e^{(J)}) - \beta_{T_e} \hat{\sigma}_{T_e}(G) \right) \right\}.$$

**Network Architecture.** We set the width of all architectures to $m = 2048$ and depth to $L = 2$. This combination is picked primarily to keep computations light, while somewhat adhering to the theoretical setup. To calculate $\hat{\sigma}_{t-1}$ we approximate the gram matrix $\boldsymbol{G}^T \boldsymbol{G}$ with its diagonal, which gets worse as the number of network parameters grow. The picked values for $m$ and $L$ producing a descriptive network, and allow us to use this diagonal approximation with a negligible cumulative error.

**Graph Neural Tangent Kernel** To implement this kernel function, we use the NTK class from the Neural Tangents library [33], and sum the base NTK via Eq. (4). This library offers the tangent kernels of every network architecture, however it is unclear how the kernel is derived for a GNN, therefore we use our own expression.

**Initialization.** We initialize the networks by directly following the definition of Eq. (2). The scaling with $1/\sqrt{m}$ is crucial in activating networks in the lazy regime. If this condition is not met, the confidence sets $[f_{\mathrm{GNN}}(\cdot; \boldsymbol{\theta}) \pm \hat{\sigma}_{t-1}(\cdot)]$ may not be valid, since $\hat{\sigma}_{t-1}$ no longer accurately describes the posterior variance of $f_{\mathrm{GNN}}$.

**Training.** When analyzing the training dynamics of $f_{\mathrm{GNN}}$, we consider SGD on the $\ell_2$-regularized loss. In practice, however, we train the network with the Adam optimizer [26] from PYTORCH [34], and without weight decay. The learning rate is set to $\eta = 0.001$. We allow $T_0 = 40$ steps of random exploration, to mimic some form of pre-training. The random exploration steps are included in our regret plots. For the first $T_1 = 100$ steps, we train the network from scratch (using the same initialization $\boldsymbol{\theta}_0$) at every step $t$, as described by the algorithm, and then in batches of $T_B = 20$ just to keep computations light. At every step $t$ we run the Adam optimizer for $J_t$ gradient descent steps, where $J_t$ is calculated via the following stopping criteria

$$J_t = \min \quad J$$

$$\text{s.t.} \quad \mathcal{L}(\boldsymbol{\theta}_{t-1}^{(J)}) \leq \mathcal{L}_0 \quad \text{or,} \quad \frac{\Delta \mathcal{L}_{t-1}^{(J-1)} - \Delta \mathcal{L}_{t-1}^{(J)}}{\Delta \mathcal{L}_{t-1}^{(J-1)}} \leq \delta_0$$

where we set $\mathcal{L}_0 = 10^{-4}$, $\delta_0 = 10^{-3}$, and

$$\Delta \mathcal{L}_{t-1}^{(J)} := \mathcal{L}(\boldsymbol{\theta}_{t-1}^{(J)}) - \mathcal{L}(\boldsymbol{\theta}_{t-1}^{(J-1)}).$$

The above criterion targets both value of the loss function and the rate at which it is decaying. Effectively, this rule stops training if either the loss is lower than a threshold $\mathcal{L}_0$, or if the loss has plateaued, i.e. the relative change in the the loss is lower than a threshold $\delta_0$. Roughly put, the two conditions on value and decay of the loss, cause the training algorithm to run longer for larger $t$ and prevent over-fitting when $t$ is small. The hyperparameters of the optimizer, i.e., $\eta, \delta_0, \mathcal{L}_0, T_B, T_0$, and $T_1$ are selected by hand and not automatically tuned.

### D.3 GNN-UCB & NN-UCB

In Section 5, we compare GNN-PE with NN-PE, GNN-UCB and NN-UCB as baselines. The pseudo-code is laid out in Algorithm 3 and Algorithm 4.

---

**Algorithm 3:** GNN-UCB

---

**Input:** $m$, $J$, $\eta$, $\lambda$, $\beta_t$, $T$

**Initialize** *network parameters to a random* $\boldsymbol{\theta}^0$, *and* $\hat{\boldsymbol{K}}_0 = \sigma^2 \boldsymbol{I}$.

**for** $t = 1 \cdots T$ **do**
   **for** $G \in \mathcal{G}$ **do**
      $\hat{\sigma}_{t-1}^2(G) \leftarrow \boldsymbol{g}_{\mathrm{GNN}}^T(G; \boldsymbol{\theta}^0) \hat{\boldsymbol{K}}_{t-1}^{-1} \boldsymbol{g}_{\mathrm{GNN}}(G; \boldsymbol{\theta}^0)/m$
      $U_{G,t} \leftarrow f_{\mathrm{GNN}}(G; \boldsymbol{\theta}_{t-1}^{(J)}) + \beta_t \hat{\sigma}_{t-1}(G)$
   **end**
   $G_t = \arg\max_{G \in \mathcal{G}} U_{G,t}$
   Select $G_t$ and append the rewards vector $\boldsymbol{y}_t$ by the observed reward.
   Set $\hat{\boldsymbol{K}}_t \leftarrow \lambda \boldsymbol{I} + \sum_{i \leq t} \boldsymbol{g}_{\mathrm{GNN}}(G_i; \boldsymbol{\theta}^0) \boldsymbol{g}_{\mathrm{GNN}}^T(G_i; \boldsymbol{\theta}^0)/mt$
   Calculate $\boldsymbol{\theta}_t^{(J)} = \mathrm{TrainGNN}\left(m, J, \eta, \lambda, \boldsymbol{\theta}^0, (G_i, y_i)_{i \leq t}\right)$
**end**

---

 

---

**Algorithm 4:** NN-UCB

---

**Input:** $m$, $J$, $\eta$, $\lambda$, $\beta_t$, $T$

**Initialize** *network parameters to a random* $\boldsymbol{\theta}^0$, *and* $\hat{\boldsymbol{K}}_0 = \sigma^2 \boldsymbol{I}$.

**for** $t = 1 \cdots T$ **do**
   **for** $G \in \mathcal{G}$ **do**
      $\hat{\sigma}_{t-1}^2(G) \leftarrow \boldsymbol{g}_{\mathrm{NN}}^T(\bar{\boldsymbol{h}}_G; \boldsymbol{\theta}^0) \hat{\boldsymbol{K}}_{t-1}^{-1} \boldsymbol{g}_{\mathrm{NN}}(\bar{\boldsymbol{h}}_G; \boldsymbol{\theta}^0)/m$
      $U_{G,t} \leftarrow f_{\mathrm{NN}}(\bar{\boldsymbol{h}}_G; \boldsymbol{\theta}_{t-1}^{(J)}) + \beta_t \hat{\sigma}_{t-1}(G)$
   **end**
   $G_t = \arg\max_{G \in \mathcal{G}} U_{G,t}$
   Select $G_t$ and append the rewards vector $\boldsymbol{y}_t$ by the observed reward.
   Set $\hat{\boldsymbol{K}}_t \leftarrow \lambda \boldsymbol{I} + \sum_{i \leq t} \boldsymbol{g}_{\mathrm{NN}}(\bar{\boldsymbol{h}}_{G_i}; \boldsymbol{\theta}^0) \boldsymbol{g}_{\mathrm{NN}}^T(\bar{\boldsymbol{h}}_{G_i}; \boldsymbol{\theta}^0)/mt$
   Calculate $\boldsymbol{\theta}_t^{(J)} = \mathrm{TrainNN}\left(m, J, \eta, \lambda, \boldsymbol{\theta}^0, (\bar{\boldsymbol{h}}_{G_i}, y_i)_{i \leq t}\right)$
**end**

---

 

---

**Algorithm 5:** TrainNN

---

**Input:** $m$, $J$, $\eta$, $\lambda$, $\boldsymbol{\theta}^0$, $(\bar{\boldsymbol{h}}_{G_i}, y_i)_{i < t}$

Define $\mathcal{L}(\boldsymbol{\theta}) = \frac{1}{t} \sum_{i < t} \left(f_{\mathrm{NN}}(\bar{\boldsymbol{h}}_{G_i}; \boldsymbol{\theta}) - y_i\right)^2 + m\lambda \|\boldsymbol{\theta} - \boldsymbol{\theta}^0\|_2^2$

Initialize $\boldsymbol{\theta}^{(0)} = \boldsymbol{\theta}^0$

**for** $j = 1, \ldots, J$ **do**
   $\boldsymbol{\theta}^{(j)} = \boldsymbol{\theta}^{(j-1)} - \eta \nabla \mathcal{L}(\boldsymbol{\theta}^{(j-1)})$
**end**

**Output:** $\boldsymbol{\theta}^{(J)}$

---

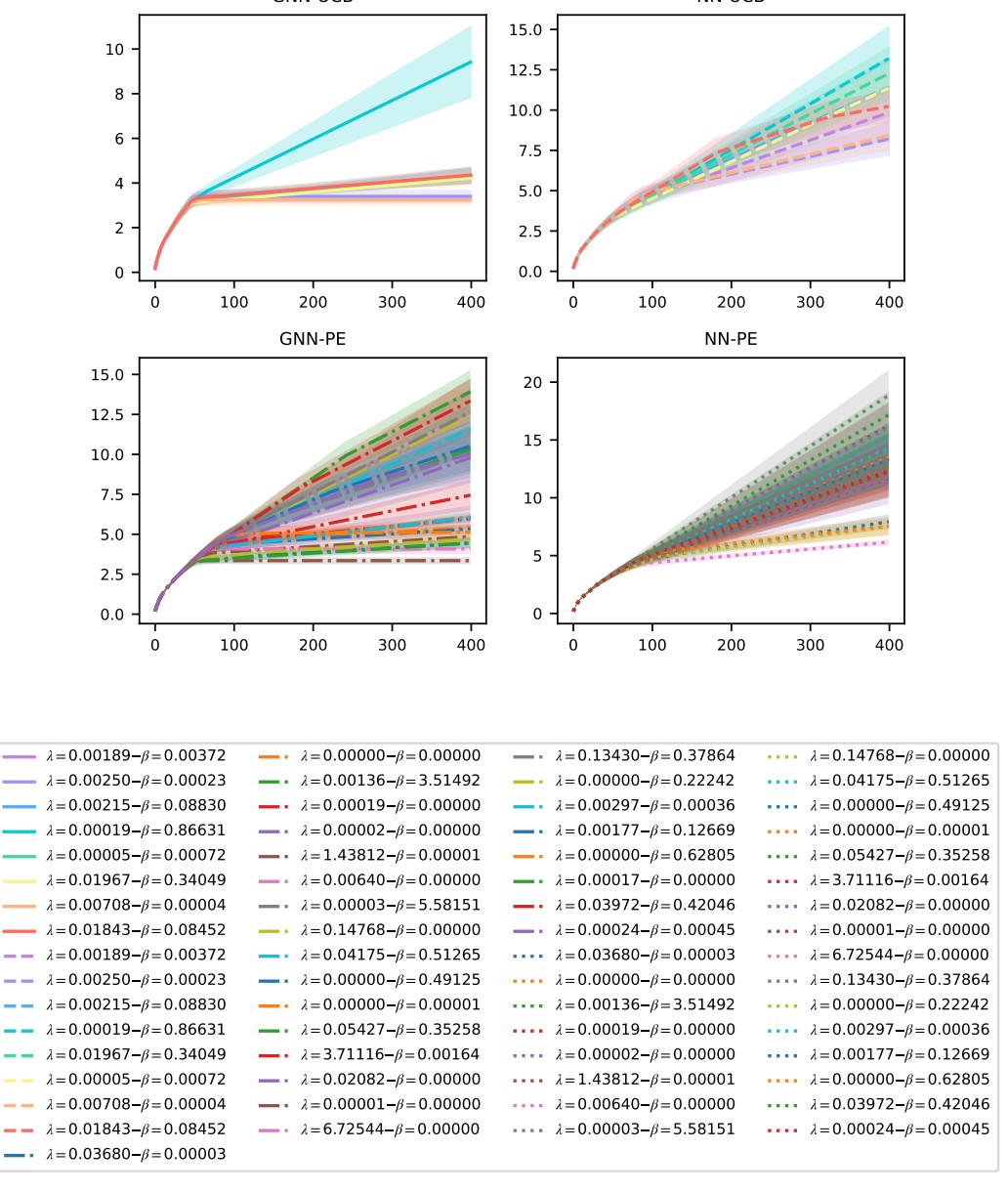

Figure 5: Results of hyper-parameter search for all algorithms. The GNN methods then to perform well for many configurations of $\lambda$ and $\beta$.

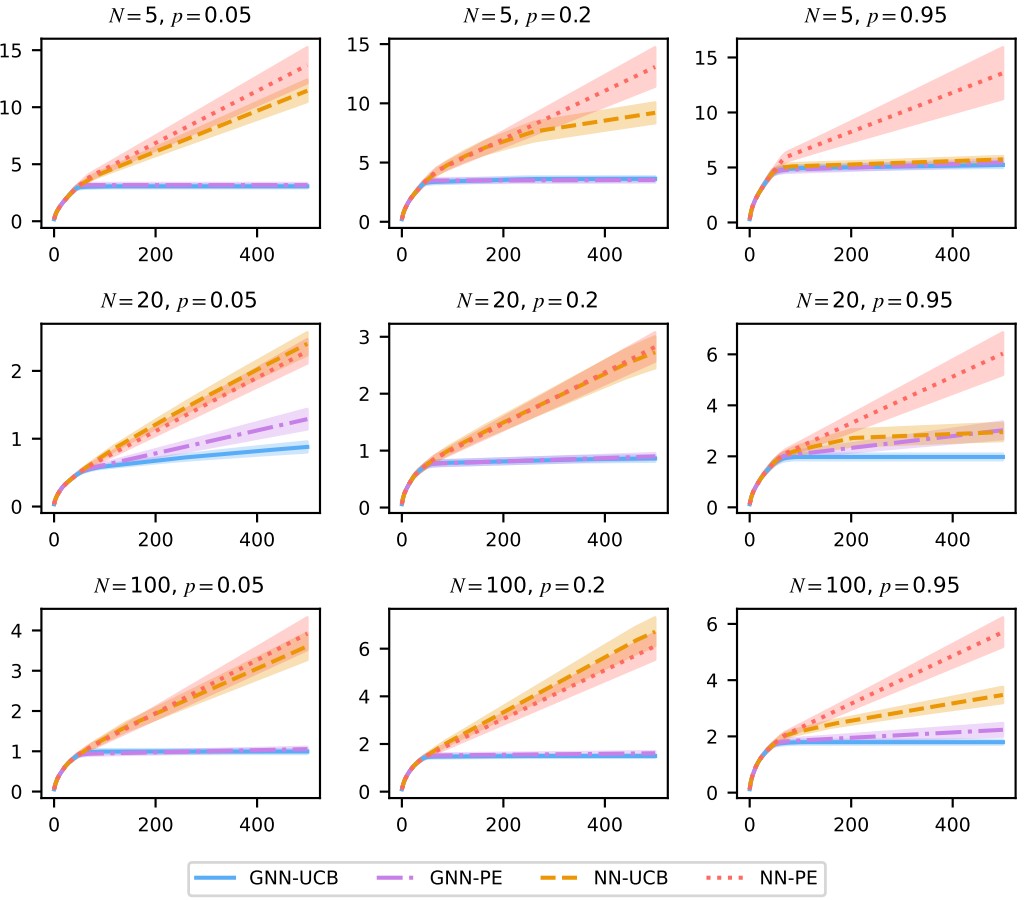

Figure 6: Comparing performance of GNN-PE, GNN-UCB, NN-PE, and NN-UCB for all dataset configurations. The GNN methods consistently outperform NN methods. Hyper-parameter tuning is done only for $N = 5, p = 0.05$ and the same is used across all setting.

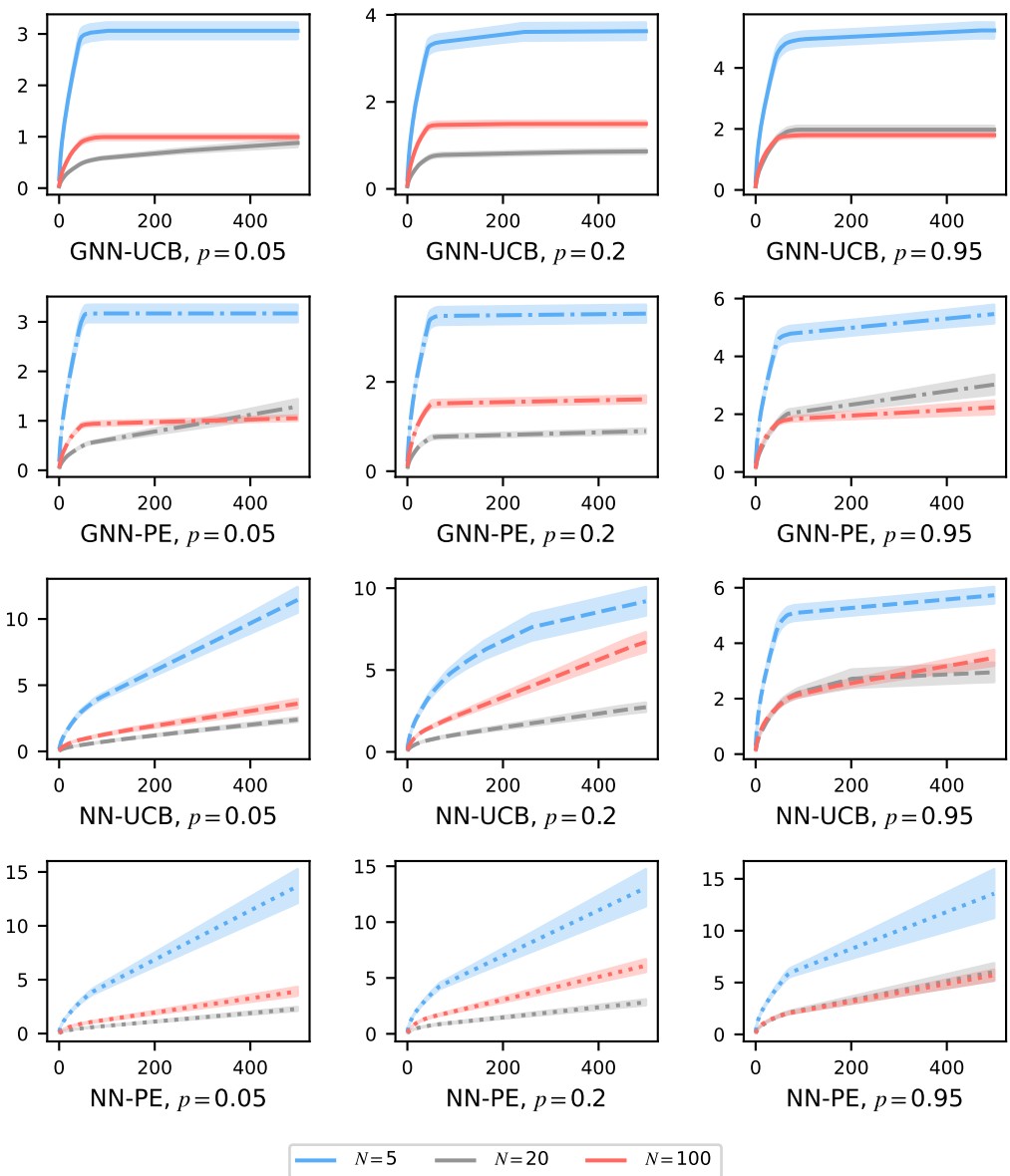

Figure 7: Effect of graph size on performance of GNN-PE, GNN-UCB, NN-PE, and NN-UCB. GNN methods perform well regardless of value of $N$. Inference on sparse small graphs is challenging since the random graphs tend to have very few edges.

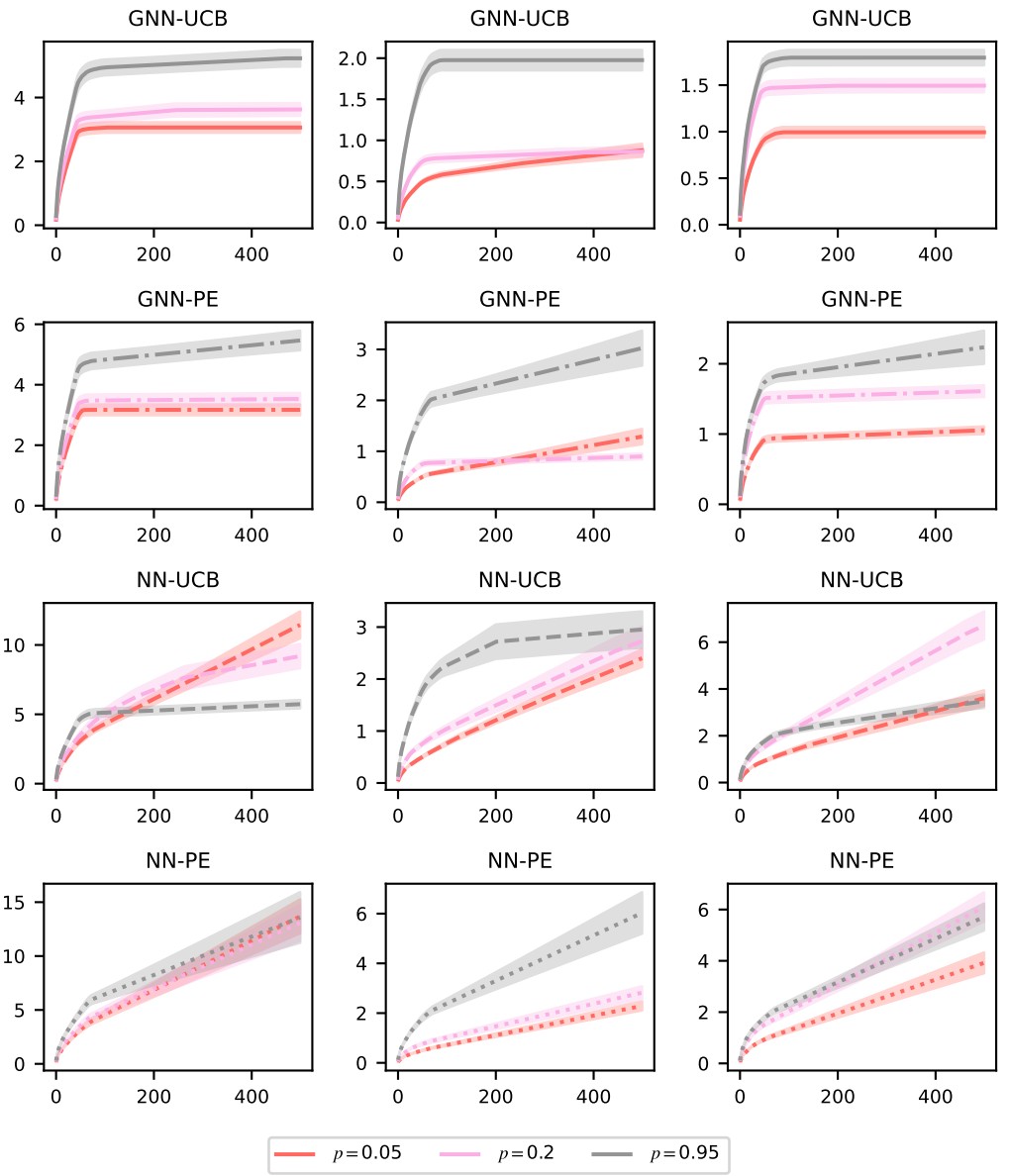

Figure 8: Effect of edge density on performance of GNN-PE, GNN-UCB, NN-PE, and NN-UCB. The NN algorithms tend to improve as $p$ grows.