# OpenReview forum: "Graph Neural Network Bandits"
_NeurIPS.cc/2022/Conference — NeurIPS 2022 Accept_

### Official Review · Reviewer_FKpp · 2022-07-11

**Rating:** 5
**Confidence:** 4
**Soundness:** 4 excellent
**Presentation:** 3 good
**Contribution:** 2 fair

**Summary:**

This paper contributes to the bandits literature wirh GNNs as reward functions:
 * "In this paper, we consider bandit optimization over graphs and propose to employ graph neural
networks to estimate the unknown reward. "
* "Without exploiting structure, standard bandit algorithms (e.g., [3]) cannot generalize across graphs"
* "Consider functions that are invariant to node permutations"
* "In this work, we use graph neural networks to estimate the unknown reward function f∗. "

·GNNs can be seen as averaged NNs each one taking as input its aggregation.
Broadly speaking, we have that the "average of local aggregations leads to the global one". This is consistent with an averaged readout.

Main result:
"Our main result are GNN confidence bounds that can be readily used in sequential
decision-making algorithms to achieve sublinear regret bounds "

GNN confidence bounds:
* Independence of N in the MIG bounds. Using GNNs vs NNs
* Gain wrt to structure-agnostic bandits.

Experiments.
* Comparison with NNs. A nice way of checking up the impact of MIG bounds.
* With dense graphs, they obtain a similar performance for both GNNs and NNs, There is a nice explanation for this in terms of message passing: similar averaged aggregations for all nodes. GNNs suffer from over-smoothing.

**Questions:**

* How the bandit optimization leads to a better understanding of the GNNs?
* As the variance in eq. 8 is defined wrt to the initial gradient, it seems that the bound is very large, please clarify.
* How the performance of the GNNs decays with more layers?

**Ethics Review Area:**

["I don’t know"]

**Limitations:**

Suggestions for improvement. It shoulod be very interesting to use all this theory for better understanding how GNNs decay in performance (beyond the dense case) and how to take rewiring decisions. For instance, the sequence of graphs could be different strategies of rewiring or different rewiring degrees and bandits could obtain the optimal parameterization and the MIG and confidence bounds.

**Strengths And Weaknesses:**

Stengths.
* Mostly for the bandits literature, theorerical insights in using GNNs as permutation invariant reward functions.
* Theoretical insignt in how to improve bandits in structural domains.

Weaknesses.
* No insights in common problems in GNNs: over-smoothing, over-squashing etc. As commented above, with dense graphs we have that GNNs and NNs bandits have a similar performance. This line of analyisis is more interesting.
* Limited interest or impact but in the bandits area.

---

> ### Author Response · Authors · 2022-08-01
> **Response to Reviewer FKpp**
>
> We thank the reviewer for their comments. Before we respond to the specific questions, we would like to highlight the main area of this work’s contribution:
>
> This paper primarily focuses on costly optimization problems on graphs when the objective function is unknown, as in drug discovery or molecule design. This literature lacks algorithms with theoretical guarantees that simultaneously achieve competitive performance. Our work is the first to fill this gap, by giving the first GNN-based solution to such problems while achieving rigorous theoretical guarantees. Consequently, we believe that our work is of significant interest to the bandit community, and can facilitate applied research on, e.g., chemical design.
>
> If this response does not lift the reviewer’s concerns regarding limited interest/impact, we would appreciate a further explanation.
>
>
>
> **Response to comments and questions**
>
> > “How does bandit optimization lead to a better understanding of the GNNs?”
>
> We actually do not consider bandit optimization for the sake of better understanding of GNNs. Conversely, we use GNNs, to design algorithms for bandit tasks. We demonstrate properties of GNNs which help us achieve competitive bandit regret bounds that scale well with the number of graph nodes.
>
> ---
>
> > “As the variance in eq. 8 is defined wrt to the initial gradient, it seems that the bound is very large.”
>
> Could you please clarify which bound you are referring to? \
> The gradients in Eq. 8 are normalized by the square root of width $\sqrt{m}$. Further, Eq. 8 contains the “inverse” matrix term in the middle which also depends on the gradients at initialization. Since the weights are normal Gaussians, this variance will be bounded by one with high probability. Empirically, we observe that this is always the case.
>
> ---
>
> > “How does the performance of the GNNs decay with more layers?”
>
> In the lazy regime, the expressive power of the network is not significantly affected by the number of layers. This is in fact one of the disadvantages of working in this regime. However, this regime gives powerful theoretical tools, hence our analysis is limited to it. We performed a small hyper-parameter search for $L$ among the values $\{2, 5, 10\}$ for our experiments. We did not observe a significant change in bandit regret performance based on the number of layers, again, since we operate in the lazy regime.
>
> ---
>
> > “Suggestions for improvement: … use all this theory for better understanding how GNNs decay in performance … and how to take rewiring decisions.”
>
> We are glad that our work has already inspired new ideas. We don't think that the existence of follow-ups should be considered a limitation of this work.
>
> We highlight that the focus of this paper is primarily on structured bandit tasks and *not* on understanding the behavior of GNNs. Our work takes the first step in the direction of utilizing GNNs for bandit optimization on graphs. However, the underlying analyses can ignite theoretical follow-ups in adjacent fields.
>
> Having addressed all of the questions provided by the reviewer, and given the contributions of this paper to the bandit literature, we kindly ask the reviewer to reconsider the assessment of our paper. We would be happy to answer any remaining questions or concerns.

---

> > ### Comment · Reviewer_FKpp · 2022-08-09
> > **Upgrade my rating**
> >
> > Based on the satisfactory answers of the authors I upgrade my rating to "bordeline-accept". There are still elemnts of GNNs that need to be considered for using them as structural bandits. My question “How does bandit optimization lead to a better understanding of the GNNs?” is answered saying that they are doing the opposite. Im aware of that but one have to consider the limitations of the GNNs before using them and after doing that it is scientifically expected to reson on the lesson learnt from GNNs.

---

> > > ### Author Response · Authors · 2022-08-09
> > > **Response to upgraded rating**
> > >
> > > Thank you for reconsidering your assessment.
> > > Could you please also update the score in your review to ``border-line accept”?
> > >
> > > We agree that as future work, one could study the implications of our results towards a better understanding of GNNs.

---

> ### Author Response · Authors · 2022-08-08
> **Reminder to Reviewer FKpp**
>
> Dear reviewer,
>
> The author-reviewer discussion period is ending in less than two days. We kindly request your acknowledgment of our responses, and that you let us know if there are any issues that you still find problematic, and/or check that your score is in agreement with your updated understanding of our work.
>
> Thank you for your time and consideration.

---

### Official Review · Reviewer_yqgD · 2022-07-13

**Rating:** 7
**Confidence:** 3
**Soundness:** 4 excellent
**Presentation:** 4 excellent
**Contribution:** 3 good

**Summary:**

This paper considers the bandit optimization problem over graph data. The key challenge is shown to be the permutation-invariance of the model, and the authors propose to use a permutation-invariant additive kernel on graphs nodes to represent the unknown reward function. The kernel is shown to be the limit of the GNNs in the NTK regime.
The achieved maximum information gain is independent of the number of nodes in the graphs (Theorem 4.1). They also study confidence sets over graph data and propose GNNs to achieve them (Theorem 4.2). Then, they use these results to get a regret bound using the GNN-PE algorithm (Theorem 4.3.). This regret is again independent of the number of nodes. The paper is then concluded with some experiments.


**Questions:**



I think this is an interesting/good paper. I just have some comments for the authors:



Major comments:

(0) Line 174-181 -- it is claimed that computing the invariant kernel with the average over all permutations is not feasible, and using GNNs can help alleviate this calculation to the feasible sum $O(N^2)$. This looks to be a "frustration of purpose" because you just want to use the kernel $\bar k$ for your analysis, and in practice, GNNs are used. So why showing that an easier computation is feasible is a contribution? (Please correct me if this is not true).  In my opinion, Proposition 3.1 and Proposition 3.2 are some great observations, not contributions because the proofs are just changes in the order of summation, etc.

(1) Can you find concrete graph functions that can be approximated with those balls in the RKHS norm (with explicit radii)? For example, node degrees, paths, functions of them, etc. I know those spaces are common in the kernel theory, but here for graphs, it is not that clear how they relate to graph properties.






Minor comments:

(0) Line 104 -- the used notation is a bit unconventional; I suggest using $S_n$ for the symmetric group and $\sigma$ for a permutation, which is more standard.

(1) Line 528 -- please add space before "consider"



**Strengths And Weaknesses:**


pros:

(1) well-motivated problem, having solid theoretical results

(2) including supporting experiments

(3) the paper is very well-written



cons:


(1) having no example about the ball in RKHS norm (see below)

---

> ### Author Response · Authors · 2022-08-01
> **Response to Reviewer yqgD**
>
> We thank the reviewer for their helpful comments. Regarding your questions:
>
> > You only use the GNTK for theoretical analysis, so why show that they are easy to compute in practice?
>
> It is true that we only use this kernel as an analytical tool. We discuss its properties because we believe that the GNTK, as a powerful graph kernel, can be of independent interest in kernel methods. Ease of computation makes it a more favorable choice.
>
> ---
>
> > “Can you find concrete graph functions that can be approximated with those balls in the RKHS norm? For example, node degrees, paths, functions of them, etc.”
>
> Our focus has been on real-valued graph functions that only take the graph as an input. So for instance, the node degree function, which takes both a graph and a node index is not directly considered in our analysis.
>
> To support such functions, one way is to alter the input domain. For instance, we can change the domain from $\mathcal{G}$ to $\mathcal{G}\times [N]$ and define a kernel on the product space, e.g., $k_{\mathrm{GNN}} \otimes k$, where $k: [N] \times [N] \rightarrow \mathbb{R}$. Another possible extension is to keep the same domain, but consider vector-valued functions $f: \mathcal{G} \rightarrow \mathbb{R}^N$, and update the tangent kernel (and in turn the RKHS) accordingly. These would be interesting extensions of the current work.
>
> In Appendix A3, we provide insights on which functions are included in the RKHS ball of the GNTK. The GNTK is a universal approximator for permutation invariant continuous functions over compact domains. All functions within the RKHS are of finite-norm and there exists $B$ such that the true reward falls into the B-ball. The standard assumption of $B$ being known, allows for constructing valid confidence bounds in the analysis of the regret bound.

---

> > ### Comment · Reviewer_yqgD · 2022-08-03
> > **Response to the authors**
> >
> > I want to thank the authors for their response. Here are some suggestions:
> >
> > (0) "You only...in practice?"
> >
> > I appreciate the authors' response to this concern. It is quite important to emphasize those facts in the paper, and I strongly suggest mentioning that (1) you only use them for your analysis, and (2) but some computation benefits are observed and proved in the paper (of potential independent interest).
> >
> >
> > (1) "Can you.., etc."
> >
> > I understand what you mentioned. Your theory suggests the possibility of approximating anything with sufficiently large $B$. I also read Appendix A3 (as you mentioned). Do you have some insights about how large should $B$ get in order to approximate some natural graph properties (like average degree, max-degree, the number of paths with a specific length in the graph, etc.)? Also, I guess it is good to mention that approximating some functions on the graph can be computationally hard, and how is this related to $B$?

---

> > > ### Author Response · Authors · 2022-08-08
> > > **Follow-up Response**
> > >
> > > Thank you for your response and your suggestions!
> > >
> > > - (0) We updated the text following your suggestion. In Section 3, we make it clear that we study properties of the GNTK for the sake of our analysis, and explain how these properties may be of independent interest.
> > >
> > > - (1) We note that our focus is on obtaining a general bound that applies to the entire class of functions with bounded norm $B$. Hence, we don’t analyze specific instances.
> > >
> > >   The answer to the reviewer’s question will depend on the concrete graph function, the kernel used in the norm definition, and properties of the graph domain (e.g. $N$ and $p$). We note that this RKHS norm can be approximated (from below) by taking the kernel integral operator for finitely many eigenfunctions of the kernel.
> > >
> > >   We have not investigated this complex question, since our primary focus is on obtaining the worst-case bounds over an entire family of B-smooth functions.  The instance-dependent analysis is out of the main scope of our work. Nevertheless, we agree with the reviewer that this is an interesting question.  Lastly, it is worth mentioning that in practice, the algorithm seems to be robust to the choice of $B$.

---

### Official Review · Reviewer_uFWG · 2022-07-20

**Rating:** 7
**Confidence:** 3
**Soundness:** 3 good
**Presentation:** 3 good
**Contribution:** 3 good

**Summary:**

In this paper, the authors address the bandit optimization problem with the reward function defined over graph domains.
Given that in applications such as molecule design and drug discovery the reward is naturally invariant to graph permutations and to be able to scale to large graph domains, the authors consider the permutation invariance in their model. Authors, extend the methods for kernelized bandits that rely on kernelized confidence sets, to optimization over graph domains and constructing confidence sets that can quantify the uncertainty of graph neural networks (GNNs) estimates. To be more precise, authors proposed a permutation invariant additive kernel, and established a connection between such kernels and the graph neural tangent kernel that are introduced in this work. Establishing this connection led to the following main contributions:
 i - establishing an upper bound for maximum information gain.
ii- constructing valid confidence sets that utilize GNNs.
iii- Leveraging the GNN-confidence bounds to propose an algorithm namely, GNN-PE (GNN-Phase Elimination) to achieve sublinear regret.




**Questions:**

This paper was assigned to me very recently. Unfortunately I did not get the chance to read the supplementary document in full, but I tried to skim through the supplementary document to verify some of the results.

- In proposition 3.1. authors show that in the infinite width limit, the tangent kernel converges to a deterministic kernel that is the graph neural tangent kernel. How limiting this condition is? Do the conditions on $m$ in the statement of theorems 4.2. and 4.3. obey this condition and the connection between the GNTK and the  permutation invariant additive kernel holds?
Also, in the statement of the proposition 3.1. I believe it should be "the tangent kernel $\tilde{k}_{GNN}$ as opposed to $k_{NN}$.
- In section 5, the numerical experiments show that the GNN-based techniques are superior to NN based methods. Is this because the GNTK inherits the properties of the permutation invariant additive kernel and this is not the case for NTK?

- I suggest the authors, also highlight the limitations of the proposed method.

Minor comments:
Please improve the figures in section 5, for example add labels.
Typo: line 534, "tangent".



**Limitations:**

The paper is more toward theoretical extension of bandit problem to the graph domain. However, authors did not clearly explain if there are limitations to their results and the assumptions they made to derive the main results.

**Strengths And Weaknesses:**

-The novelty of the paper lies in establishing a connection between the permutation invariant additive kernels and the graph neural tangent kernel (GNTK) that authors introduced in this work. In addition authors linked the GNTK to neural tangent kernel (NTK). These results are shown in propositions 3.1 and 3.2. and are the basis for the main contributions of the paper. Leveraging the notion of GNTK, the authors calculate the information gain of a sequence of actions and the maximum information gain (MIG). Also, they obtained a bound for MIG that does not depend on N that is the number of nodes in the graph. In addition, authors proposed valid confidence sets for GNN and employ them to develop an algorithm to solve the bandit optimization tasks on graphs.
-Authors provided proofs to the discussed theorems and propositions in the supplementary document.

---

> ### Author Response · Authors · 2022-08-01
> **Response to Reviewer uFWG**
>
> We thank the reviewer for their helpful comments. We incorporated your feedback regarding the presentation; added labels and made the figures more readable. Regarding your questions:
>
> > Clarification regarding the conditions on width of the network.
>
> *Neither* of our theorems assume that the network is of infinite width. Theorem 4.2 and 4.3 are statements about uncertainty and regret bounds that are achieved with *finite width* GNNs, where the width $m$ depends polynomially on the problem’s parameters.
>
> ---
>
> > “In section 5, the numerical experiments show that the GNN-based techniques are superior to NN based methods. Is this because the GNTK inherits the properties of the permutation invariant additive kernel and this is not the case for NTK?”
>
> Yes. A permutation invariant additive reward estimator (i.e, the GNN) exploits the structural symmetries that are inherent to the graph bandit problem, and achieves a better performance.
>
> ---
>
> > “I suggest the authors also highlight the limitations of the proposed method.”
>
> In Section 2, we state the setting under which our proposed method works. Following your suggestion, we made it more visible that our result supports GNNs with a single convolutional layer.

---

### Meta-Review · Area_Chair_7e9z · 2022-08-27

**Recommendation:** Accept
**Confidence:** Certain

**Metareview:**

This paper studies a bandit optimization problem where the rewards are smooth on a graph. The authors show that GNNs can be used to estimate the reward functions and confidence bounds. This is then used to design a phased elimination algorithm. The reviewers agree that the paper is interesting and makes a significant contribution to the area -- I recommend its acceptance.

**Award:**

No

---

### Decision · Program_Chairs · 2022-09-14

Accept